# Vaccine hesitancy among paediatric nurses: Prevalence and associated factors

**Usue Elizondo-Alzola**[1,2,3]*, **Mireia G. Carrasco**[1,3], **Laia Pinós**[4], **Camila Andrea Picchio**[1,5], **Cristina Rius**[1,3,6,7], **Elia Diez**[1,3,6,7]

**1** Barcelona Public Health Agency, Barcelona, Spain, **2** Pompeu Fabra University, Barcelona, Spain, **3** Autonomous University of Barcelona, Barcelona, Spain, **4** Preventive Medicine and Epidemiology Department, Vall d'Hebron Barcelona Hospital Campus, Barcelona, Spain, **5** Barcelona Institute for Global Health (ISGlobal), Hospital Clínic, University of Barcelona, Barcelona, Spain, **6** Biomedical Research Institute Sant Pau (IIB Sant Pau), Barcelona, Spain, **7** CIBER Epidemiology and Public Health (CIBERESP), Carlos III Institute, Madrid, Spain

☯ These authors contributed equally to this work.
* usue.elizondo@gmail.com

**Data Availability Statement:** The database file is available from the Figshare database: https://figshare.com/s/f0257774983e224a9293.

**Funding:** The authors received no specific funding for this work.

## Abstract

### Objective

This study describes the prevalence of vaccine hesitancy associated with the Catalan systematic childhood vaccination calendar and some related psychosocial determinants among paediatric primary care nurses in Barcelona (Spain).

### Methods

Cross-sectional descriptive study. In 2017 we invited the paediatric nurses (N = 165) working in Barcelona public primary health centres with paediatric departments (N = 41) to participate. They answered a questionnaire with sociodemographic and behavioural variables: severity and perceived probability of contracting the diseases of the vaccines in the vaccination schedule; safety and protection offered by each vaccine; and beliefs, social norms, and knowledge about vaccines. Outcome variable was vaccine hesitancy, dichotomized into not hesitant (nurses who would vaccinate their own offspring), and hesitant (including those who would not vaccinate them, those who had doubts and those who would delay the administration of one or more vaccines). We performed bivariate analysis and adjusted logistic regression models.

### Results

83% of paediatric nurses (N = 137) agreed to participate. 67.9% had the intention to vaccinate their children of all the vaccines in the systematic schedule. 32.1% of nurses experienced vaccine hesitancy, especially about the HPV (21.9%) and varicella (17.5%) vaccines. The multivariate analysis suggests associations between hesitancy and low perception of the severity of whooping cough (aOR: 3.88; 95%CI:1.32–11.4), low perception of safety of the HPV vaccine (aOR:8.5;95%CI:1.24–57.8), the belief that vaccines are administered too early (aOR:6.09;95%CI:1.98–18.8), and not having children (aOR:4.05;95%CI:1.22–13.3).

**Competing interests:** The authors have declared that no competing interests exist.

## Conclusions

Although most paediatric nurses had the intention to vaccinate their own children, almost one-third reported some kind of vaccine hesitancy, mainly related to doubts about HPV and varicella vaccines, as well as some misconceptions. These factors should be addressed to enhance nurses' fundamental role in promoting vaccination to families.

## Introduction

Vaccination is a demonstrably effective, safe, and cost-effective intervention [1]. However, in several high-income European countries, such as Italy and France, immunization rates of some immuno-preventable diseases such as measles have declined in recent decades, which has contributed to recent outbreaks of the measles disease [2]. Vaccine Hesitancy (VH) could in part be responsible for this growing global phenomenon. VH is defined as the reluctance or refusal to vaccinate despite the availability of vaccines, threatens to reverse progress made in tackling immuno-preventable diseases [3]. In 2019, the World Health Organization (WHO) placed VH among the top 10 threats to global health [4].

VH is complex and specific for each context and type of vaccine. In Europe, some common reasons against vaccination include the lack of confidence in vaccines, in their administration, in the public health services, and in the pharmaceutical industry [5]. In the last decade, social networks and some digital media have contributed to expanding these doubts and to eroding families' trust in healthcare professionals (HCPs) [6, 7]. In the European Union, around 20% of parents report having doubts about vaccinating their children [8]. In France, 36% of parents question the safety of vaccines [9], and in Spain, controversies are quite similar to those in other countries: for example, the false association between the MMR vaccine and autism is not uncommon; as well as the belief that the pharmaceutical industry influences the public vaccination schedule [10]. Moreover, 8% of Spanish people think that vaccination carries more risks than benefits [11].

Health care professionals still maintain a positive influence on family's decisions to vaccinate their children [12]. In Spain, 69% of families reported that paediatricians were their most important source of information [13]. In this country, paediatric primary health care (PHC) teams include medical and nursing HCPs. PHC nurses can administer vaccines without a medical prescription [14]. All vaccines included in the official schedule are recommended, not required, so positive communication between families and nurses is crucial to maintain high vaccination coverage rates [15]. Paediatric nurses use an important part of their consultation time for tasks related to vaccination, except in complex non-routine cases, which are attended by paediatricians. These functions, together with their accessibility, make nurses a key actor in the vaccination process [16].

A study among paediatric PHC teams in Barcelona found that 25% of the HCPs involved in vaccination had doubts about at least one of the vaccines on the systematic vaccination schedule, especially the human papillomavirus vaccine. The authors found that up to 40% of the respondents lacked specific knowledge about vaccination. The study found some differences between paediatricians and paediatric nurses and nurses reported more VH (adjusted odds ratio: 2.0; 95% CI: 1.1–3.7), and recommended exploring this phenomenon in greater detail [17].

Given the ongoing COVID-19 pandemic and the need to roll-out mass vaccination campaigns, vaccine acceptance by HCPs is crucial. A dip in vaccine confidence among HCPs

could impact implementation for the general population who trust HCPs for their vaccine information and recommendations. In a recent systematic review of COVID-19 vaccine acceptance rates among HCPs (including physicians and nurses), COVID-19 vaccine acceptance rates ranged from 27.7% in the Democratic Republic of the Congo to 78.1% among physicians in Israel, where the rate among nurses surveyed was lower (61.1%) [18]. Two studies that dated back to the earlier part of the pandemic (2020 February and March) among nurses in Hong Kong also reported low rates of COVID-19 vaccination acceptance (40.0% and 63.0%) [19, 20]. A relevant increase in the acceptance of COVID-19 vaccines among HCP was observed according to a survey; and suggests that the increased intention to receive a vaccine could be influenced by the transparent development process [21].

The aim of the current study was to determine the prevalence of VH in regards to the vaccines in the systematic childhood vaccination calendar of Catalonia (Spain) and to explore some of its psychosocial determinants among PHC paediatric nurses in the city of Barcelona.

## Materials and methods

The study 2018/7790/I was approved by the ethics committee of the CEIm-Parc de Salut MAR of Barcelona. The study was conducted in accordance with the principles of the Declaration of Helsinki. All participants in the survey provided signed informed consent.

### Study design and participants

We conducted a cross-sectional study. The study population included all the paediatric nurses working in the 41 public PHC centres with paediatric departments in the city of Barcelona in 2017. We included nurses who performed care with patients (not nurses working in management) and excluded students, residents (because they are in a training period), and temporary nurses (because of their short working periods in the centre). The population sample was 165 nurses.

### Data collection

We collected the information using a questionnaire-based in literature [6, 22, 23] (S1 Appendix), translated into Catalan and Spanish and culturally adapted using the cognitive debriefing method by the research team [24]. Cognitive debriefing is a process where representatives of the target population actively test the translated questionnaires to determine whether respondents would understand the questionnaire as easily as the primary version would be understood [25]. The self-administered questionnaires were made available to nursing staff between March 2016 and February 2017. The questionnaires were given on paper to the participating nurses. After an initial email sent out to all PHC centres, an 11-month period was needed to schedule the visits on days and times suitable for all. Additionally, PHC centres who did not initially respond were followed-up up to 6 times via email in order to include them in the study. In reaching out to each centre referent and speaking with them on a personal level, researchers were able to contact and reach almost all PHC centres (only one was excluded) with paediatric departments in the public health system in the city of Barcelona. Two researchers attended the PHC centre and after a brief explanation of the study objectives distributed the questionnaire for participants to respond on their own. Finally, the participants answer the questionnaires and had however much time they needed to complete them, but most completed it in about 15 minutes. Data was entered using TeleForm® software. Any errors detected were compared with the original survey and changed. This data was imported to Stata version 12.0 (*Stata Corp. 2011. Stata Statistical Software*: *Release 12. College Station*, *TX*: *StataCorp LP)* by one researcher who cleaned the data in this software.

The questionnaire (S2 Appendix) gathered sociodemographic information (age, sex, years of experience, offspring), and psychosocial determinants based on theoretical models of behaviour: intention to vaccinate their own children, self-efficacy about answering family questions, perception of the severity and probability of contracting the "immune-preventable diseases", safety and protection conferred by vaccines in the systematic childhood vaccination schedule in Catalonia [16], beliefs, social norms, and knowledge about vaccines, as well as myths and doubts posed to nurses by the families [26–28].

## Variables

The outcome variable was VH, a dichotomous variable constructed from the variable on the intention to vaccinate their offspring for each of the 14 vaccines in the vaccination calendar. We used the intention to vaccinate based on the Theory of Reasoned Action which is a theory of planned behaviour, and the integrated behavioural model [29]. The question was: "If you had a child today, would you agree for them to receive the vaccines in the current systematic schedule?" We coded: "vaccine hesitancy" if they responded either; "no", "I have doubts" or "I would do it later" for one or more vaccines, and "non-hesitancy" if they responded that they would vaccinate according to the schedule.

In line with the Health Belief Model [30], and relying on the results and recommendations of a systematic review which employed this theory among HCPs [31], we collected data on the participants' perception of the severity and probability of contracting each immuno-preventable disease, and the safety and protection conferred for each vaccine in the schedule. We collected the answers on a 5-point Likert scale in addition to a "do not know/no response" option. Then, we created dichotomous variables, excluding non-responses as follows: probability of contracting the disease, "probable/very probable" vs other responses; severity of the illness, "serious/very serious" vs other responses; safety of the vaccine, "safe/very safe/totally safe" vs other responses; and protection conferred by the vaccine, "protective/totally protective" vs other responses. The perception of the severity of HPV infection was not included in the severity section because the question referred to 8-year-old girls or boys and it is understood that at childhood they cannot become infected by this virus nor suffer from cervical cancer.

Answers regarding beliefs, social norms, and knowledge were collected in five categories and dichotomized into "agreement" or "disagreement" with the most favourable option to vaccination, depending on how the question was stated.

## Analysis of data

We carried out a descriptive analysis of the data. We studied the relationship between VH and explanatory variables using the chi-square or Fisher's exact tests. After verifying that the data were normally distributed, we analysed age and years of experience as continuous variables using the Student's t-test. We fitted logistic regression models using the variables statistically significant in the bivariate analysis and adjusted for sex, years of experience and offspring. The variables included in the models are described in Tables 2 and 3. We computed the adjusted odds ratios (aOR) and their 95% confidence intervals (CI). We compared the models based on the likelihood ratio test and chose the model providing the most information with the fewest variables.

We analysed "do not know/no response" (DK/NR) responses and the missing values together. Missing values accounted for less than 5%. When the percentage exceeded 5%, data were analysed by including and excluding them as a category. As missing values did not affect the results, we excluded them from the analysis. Statistical significance was set at $\alpha = 0.05$. The analysis was conducted using Stata software 13.0 *(Stata Corp. 2013. Stata Statistical Software*: *Release 13. College Station*, *TX*: *StataCorp LP)*.

## Results

Of the total number of nurses working in the PHC of the public health system (n = 165), 83% (n = 137) participated in the study. Of these, 96.4% of them were women, with a mean age 47.7 years (SD: 10.2), 72.3% had children, and had an average of 23.8 years (SD: 10.5) of professional experience (Table 1).

### Intention to vaccinate

Of the nursing staff 67.9% stated their total agreement to vaccinate their children with all the vaccines in the systematic schedule. Some (32.1%) reported having doubts, delaying the administration of certain vaccines or avoid vaccinating their children with at least one of the vaccines in the systematic schedule. Excluding the varicella and HPV vaccines, the number dropped to 16.8%. HPV (21.9%), varicella (17.5%), hepatitis A (9.4%) and pneumococcal (8.8%) vaccines generated the greatest hesitancy (Fig 1).

### Perception of risk and benefit

In 7 of the 14 diseases of the vaccination schedule the perception of the risk of infection was similar in non-hesitant nurses and hesitant nurses. Hesitant nurses had a low perception risk of infection for 5 for diphtheria, whooping cough, polio, measles and HPV. They also reported a low perception of severity for 7 of them: whooping cough, *H. influenzae b*, meningococcal disease, hepatitis A, measles, mumps, and varicella (Table 2).

More than 90% of vaccine-hesitant nurses perceived all these vaccines to be very safe, except for the HPV vaccine, which was considered to be very safe by 76.5% (n = 26) of the vaccine-hesitant nurses, and by 97.7% (n = 85) of the non-hesitant nurses (p<0.001). Hesitant nurses

**Table 1. Characteristics of the paediatric nursing population stratified by vaccine hesitancy.** Barcelona. 2016–17.

| | | | Vaccine hesitancy | | | | |
|---|---|---|---|---|---|---|---|
| | Total | | Yes[1] | | No[2] | | *p value* |
| | (N = 137) | | (N = 44) | | (N = 93) | | |
| | | | N (%) | | | | |
| **Sex** | | | | | | | |
| Male | 5 | (3.6) | 2 | (4.6) | 3 | (3.2) | 0.701 |
| Female | 132 | (96.4) | 42 | (95.5) | 90 | (96.8) | |
| **Age** | | | | | | | |
| **Mean, *years* (SD)** | 47.7 | (10.2) | 46.3 | (10.9) | 48.4 | (9.8) | |
| ≤42 years | 49 | (36) | 17 | (38.6) | 32 | (34.8) | 0.892 |
| 43–56 years | 47 | (34.6) | 15 | (34.1) | 32 | (34.8) | |
| ≥57 years | 40 | (29.4) | 12 | (27.3) | 28 | (30.4) | |
| **Professional experience** | | | | | | | |
| **Mean, *years* (SD)** | 23.8 | (10.5) | 22.9 | (12.0) | 24.3 | (9.9) | |
| ≤17 years | 43 | (33.9) | 15 | (37.5) | 28 | (32.2) | 0.698 |
| 18–30 years | 48 | (37.8) | 13 | (32.5) | 35 | (40.2) | |
| ≥31 years | 36 | (28.4) | 12 | (30.0) | 24 | (27.6) | |
| **Have children** | | | | | | | |
| No | 38 | (27.7) | 19 | (43.2) | 19 | (20.4) | 0.005 |
| Yes | 99 | (72.3) | 25 | (56.8) | 74 | (79.6) | |

[1] Yes: Participants responded "no" or "I have doubts" or "I would do it later" for at least one vaccine of the 14 listed vaccines.

[2] No: Participants responded "yes" to all vaccines.

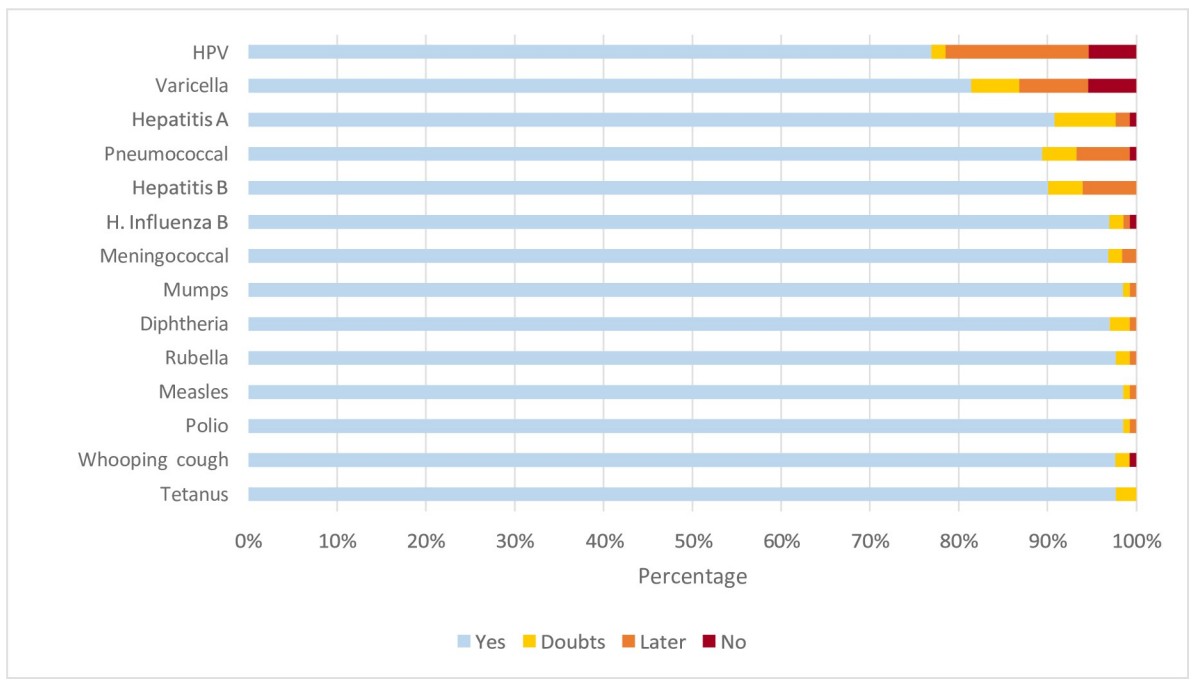

**Fig 1. Paediatric nurses responding "yes, doubts, later, or no" to vaccinating their own children (%).** Barcelona. 2016–17 (N = 137).

had a lower perception of the protection offered by the HPV vaccine, varicella vaccine, and whooping cough vaccines than those in the non-hesitant group. The perception of protection offered by the other vaccines was greater than 90% in in VH and in non-VH nurses (Table 2).

## Beliefs, knowledge, and social norms

Table 3 describes and compares beliefs, social norms and knowledge about vaccination for each group.

Hesitant nurses were more likely than non-hesitant to agree that children should only be vaccinated for serious illnesses (50.0% vs. 17.1%, p<0.001), that they receive more vaccines than necessary (52.3% vs 17.9%, p <0.001), that it is better for them to develop immunity through disease than through vaccination (39% vs 17.1%, p = 0.007), and that at least one vaccine is administered too early (70.5% vs 57.5%, p = 0.003).

Both, the hesitant and non-hesitant groups, agreed in similar proportions that the government (54.8% vs 58.3%) and the pharmaceutical industry (69.1% vs 61.5%) have illegitimate interests that influence the vaccination schedule. 100% of hesitant and 96.7% non-hesitant professionals agreed that, thanks to research, vaccines are getting better and more effective. Also, 93% of hesitant and 96.7% non-hesitant professionals agreed that vaccines are one of the safest health measures available. Moreover, almost all nurses participating agree that vaccines strengthen the immune system (88.1% VH; 92.2% non-VH)".

We did not find differences in social norms and knowledge. For the question: "At least one vaccine on the calendar contains thimerosal", missing values were 17% in the non-hesitant group and 36% in the hesitant group. Similarly, for the question: "The amount of thimerosal in vaccines causes neurotoxicity", missing values were 11.8% and 7.7% in the hesitant and non-hesitant groups, respectively. Similarly, for the question on whether vaccines contain aluminium, 31.8% of values were missing in the hesitant group and 24.7% in the non-hesitant group. Even though

**Table 2. Nurses' views on disease susceptibility and severity, and vaccine safety and protection by hesitancy Barcelona 2016–17 (N = 137).**

| | High disease susceptibility [a] | | | High disease severity [b] | | | High vaccine safety [c] | | | High vaccine protection [d] | | |
|---|---|---|---|---|---|---|---|---|---|---|---|---|
| | Vaccine hesitancy | | | Vaccine hesitancy | | | Vaccine hesitancy | | | Vaccine hesitancy | | |
| | Yes (N = 44) | No (N = 93) | *p value* | Yes (N = 44) | No (N = 93) | *p value* | Yes (N = 44) | No (N = 93) | *p value* | Yes (N = 44) | No (N = 93) | *p value* |
| **Disease or vaccine antigen** | | | | | | | | | | | | |
| **Diphtheria, %** | 34.1 | 55.2 | 0.04 | 78.6 | 89.9 | 0.643 | 100 | 100 | NC | 100 | 89.8 | NC |
| **Tetanus, %** | 47.6 | 62.5 | 0.163 | 95.1 | 95.6 | 0.912 | 100 | 100 | NC | 100 | 100 | NC |
| **Whooping cough, %** | 71.4 | 88.8 | 0.013 | 34.9 | 63.3 | 0.004 | 92.5 | 100 | NC | 68.3 | 77.5 | 0.42 |
| **Polio, %** | 14.6 | 33.0 | 0.038 | 86.0 | 92.1 | 0.242 | 100 | 100 | NC | 100 | 100 | NC |
| **H. Influenzae b, %** | 47.5 | 67.9 | 0.052 | 47.5 | 67.0 | 0.023 | 100 | 100 | NC | 94.7 | 95.5 | 1 |
| **Hepatitis B, %** | 43.9 | 60.2 | 0.124 | 62.8 | 77.5 | 0.056 | 100 | 100 | NC | 92.1 | 92.1 | 0.736 |
| **Meningococcal C, %** | 41.5 | 53.9 | 0.277 | 80.5 | 93.3 | 0.016 | 100 | 98.9 | NC | 92.3 | 98.8 | 0.085 |
| **Hepatitis A, %** | 46.3 | 57.5 | 0.327 | 19.5 | 42.2 | 0.024 | 100 | 100 | NC | 95.0 | 95.5 | 1 |
| **Measles, %** | 65.1 | 80.0 | 0.048 | 30.2 | 61.8 | 0.001 | 100 | 100 | NC | 92.7 | 95.5 | 0.675 |
| **Rubella, %** | 50.0 | 64.0 | 0.190 | 27.9 | 46.6 | 0.057 | 100 | 100 | NC | 92.7 | 97.8 | 0.169 |
| **Mumps, %** | 64.3 | 75.0 | 0.164 | 18.6 | 40.0 | 0.019 | 97.4 | 100 | 0.297 | 95.5 | 88.6 | 0.753 |
| **HPV [e], %** | 61.5 | 78.2 | 0.049 | NA | NA | NA | 76.5 | 97.7 | <0.001 | 46.9 | 80.2 | <0.001 |
| **Varicella, %** | 97.6 | 95.6 | 1 | 4.7 | 19.1 | 0.034 | 94.7 | 97.8 | 0.579 | 61.5 | 84.3 | 0.003 |
| **Pneumococcal, %** | 61.0 | 76.7 | 0.119 | 61.9 | 66.3 | 0.527 | 100 | 100 | NC | 90.0 | 93.3 | 0.492 |

NA: Not applicable; NC: Not calculable; HPV: Human papilloma virus

[a] High perception of disease susceptibility: probable and very probable.

[b] High perception of disease severity: severe and very severe.

[c] High perception of the vaccine safety: safe, very safe, totally safe.

[d] High perception of the vaccine protection: protection and significant protection.

[e] HPV severity was not considered given that the question in this block referred to an 8 year old child and it is understood that at this age they do not get the virus, because HPV that not cause an acute disease.

there were no statistical differences between both groups. In order to explore better these missing values, we checked if the participants had answered previous questions, and in general, they did.

## Factors associated with VH

The main factors associated with VH were low perception of the severity of whooping cough (aOR: 3.88; 95% CI: 1.32–11.41), low perception of the safety of the HPV vaccine (aOR: 8.50; 95% CI: 1.24–57.8), and the belief that at least one of the vaccines in the current schedule is administered too early (aOR: 6.09; 95% CI: 1.98–18.8) (Table 4). Nurses that did not have children were more likely to report hesitancy (aOR: 4.05; 95% CI:1.22–13.3).

## Discussion

Overall, the findings revealed that most nurses in PHC have a positive perception of childhood vaccination. In our study, almost 70% of the paediatric nurses reported acceptance of all the systematic childhood vaccines of the Catalan vaccination schedule in Barcelona and the remaining have questions about the administration of at least one of these vaccines. The vaccines that generated most doubts were those against HPV, varicella, pneumococcus and hepatitis A. Vaccine-hesitant nurses had a lower perception of risk caused by some diseases, a lower perception of the benefit of the varicella and HPV vaccines, and generally more unfavourable

**Table 3. Vaccine beliefs, knowledge and social norms by vaccine hesitancy in paediatric nurses.** Agreement %. Barcelona. 2016–2017 (N = 137).

| | Vaccine hesitancy | | |
|---|---|---|---|
| | Yes (N = 44) | No (N = 93) | *p* value |
| | % | % | |
| **Beliefs** | | | |
| Children should only be vaccinated for serious diseases | 50.0 | 17.1 | <0.001 |
| Children receive more vaccines than they need | 52.3 | 17.9 | <0.001 |
| I am concerned that the immune system of children may be weakened due to receiving an excessive amount of vaccines | 30.9 | 14.4 | 0.022 |
| I am more likely to trust vaccines that have been around longer than newer ones | 43.2 | 31.1 | 0.169 |
| It is better for children to develop immunity by having the illness than through vaccination | 39.0 | 17.1 | 0.007 |
| At least one of the vaccines in the current vaccination schedule is administered too early | 70.5 | 57.5 | 0.003 |
| Vaccines in the current vaccination schedule are influenced by illegitimate governmental interests | 54.8 | 58.3 | 0.703 |
| Vaccines in the current vaccination schedule are influenced by illegitimate pharmaceutical interests | 69.1 | 61.5 | 0.403 |
| Continuing to vaccinate children against Polio in Spain is acceptable even though it has been eliminated from the country[a] | 97.7 | 97.7 | 0.992 |
| Vaccines are one of the safest sanitary measures[a] | 93.0 | 96.7 | 0.344 |
| Thanks to scientific research, vaccines are increasingly better and effective[a] | 100 | 96.7 | 0.226 |
| Vaccines strengthen the immune system[a] | 88.1 | 92.2 | 0.361 |
| **Social norms** | | | |
| People in my immediate environment are in favor of vaccination | 100 | 97.8 | NC |
| **Knowledge** | | | |
| MMR vaccine can cause autism | 15.8 | 9.0 | 0.262 |
| At least one vaccine in the vaccination calendar contains thimerosal | 75.0 | 55.8 | 0.075 |
| The amount of thimerosal in vaccines can cause neurotoxicity | 51.5 | 53.9 | 0.822 |
| The amount of aluminum in vaccines can cause neurotoxicity | 45.7 | 19.0 | 0.735 |
| Having an egg allergy is a contraindication for MMR vaccine | 24.4 | 15.7 | 0.237 |
| The varicella vaccine can cause an attenuated varicella | 62.5 | 46.0 | 0.084 |
| At least one vaccine in the vaccination calendar contains aluminum | 86.7 | 74.3 | 0.171 |

NC: Not calculable. MMR: Measles, Mumps, and Rubella.

a Agreeing express the favorable option to vaccination.

beliefs about vaccination (e.g., the time of administration or the number of vaccines) than non-hesitant nurses.

Although we would like to compare our results with studies with the same aim targeting, to our knowledge, there are no other published studies with these characteristics, highlighting the research gap in studying VH in this population. We therefore put our findings in the context of studies targeting HCPs (in general) working in PHC. Compared with other European countries, Barcelona paediatric nurses appear to be less likely to have the intention to vaccinate their offspring according to the systematic childhood vaccination schedule than professionals in countries like Switzerland [22], where 95% of paediatricians would vaccinate. On the other hand, general practitioners in France are less likely to recommend vaccines to their patients than to their offspring [32]. Other authors, also in France, found that HCPs have divergent immunization attitudes toward their relatives and their patients when asking about the intention of vaccinating their own children [33], especially when considering the newest and most controversial vaccines, like the HPV vaccine. A cross-sectional study done in Croatia through a self-administered questionnaire on attitudes, beliefs and behaviours relating to vaccination among HCPs, including paediatric nurses, reported that nurses were more likely than paediatricians to be vaccine-hesitant (aOR = 5.73, 95%CI = 2.48–13.24). Therefore, in general, our results are similar to those reported across other European studies [34].

**Table 4. Factors associated to vaccine hesitancy in paediatric nurses of PHC.** Barcelona. 2016–2017.

| FACTORS | OR (95% CI) | aOR (95%CI)* |
|---|---|---|
| **ILLNESS SUSCEPTIBILITY** | | |
| **Diphtheria** | | |
| High | 1 | |
| Low | 2.37 (1.01–5.13) | |
| **Whooping cough** | | |
| High | 1 | |
| Low | 3.16 (1.24–8.08) | |
| **Polio** | | |
| High | 1 | |
| Low | 2.87 (1.08–7.60) | |
| **Measles** | | |
| High | 1 | |
| Low | 2.14 (0.95–4.83) | |
| **HPV** | | |
| High | 1 | |
| Low | 2.24 (0.98–5.09) | |
| **ILLNESS SEVERITY** | | |
| **Whooping cough** [a] | | |
| High | 1 | 1 |
| Low | 3.23 (1.50–6.89) | 3.88 (1.32–11.41) |
| *H. Influenzae B* | | |
| High | 1 | |
| Low | 2.25 (1.05–4.82) | |
| **Meningococcal disease** | | |
| High | 1 | |
| Low | 3.35 (1.08–10.41) | |
| **Hepatitis A** | | |
| High | 1 | |
| Low | 3.01 (1.25–7.25) | |
| **Measles** | | |
| High | 1 | |
| Low | 3.73 (1.71–8.13) | |
| **Mumps** | | |
| High | 1 | |
| Low | 2.92 (1.21–7.00) | |
| **Varicella** | | |
| High | 1 | |
| Low | 4.84 (1.06–22.00) | |
| **VACCINE SAFETY** | | |
| **HPV** [b] | | |
| High | 1 | 1 |
| Low | 13.08 (2.61–65.46) | 8.50 (1.24–57.80) |
| **VACCINE EFFECTIVENESS** | | |
| **HPV** | | |
| High | 1 | |
| Low | 4.60 (1.92–11.02) | |
| **Varicella** | | |

(*Continued*)

**Table 4.** (Continued)

| FACTORS | OR (95% CI) | aOR (95%CI)* |
|---|---|---|
| High | 1 | |
| Low | 3.34 (1.41–7.92) | |
| **VACINE RELATED BELIEFS** | | |
| **Children should only be vaccinated for serious illnesses** | | |
| Disagree | 1 | |
| Agree | 4.30 (1.77–10.45) | |
| **Children receive more vaccines than they need** | | |
| Disagree | 1 | |
| Agree | 2.98 (1.08–8.20) | |
| **The immune system of children may be weakened due to receiving an excessive amount of vaccines** | | |
| Disagree | 1 | |
| Agree | 5.80 (1.42–23.7) | |
| **It is better for children to develop immunity by having the illness that through vaccination** | | |
| Disagree | 1 | |
| Agree | 2.27 (0.54–9.57) | |
| **At least one of the vaccines in the calendar is administered too early [c]** | | |
| Disagree | 1 | 1 |
| Agree | 6.00 (2.28–15.80) | 6.09 (1.98–18.77) |
| **SOCIODEMOGRAPHIC CHARACTERISTICS** | | |
| **Sex** | | |
| Female | 1 | |
| Male | 1.42 (0.23–8.90) | |
| **Professional experience** | | |
| <17 years | 1 | |
| 18–30 years | 0.69 (0.28–1.70) | |
| >31 years | 0.93 (0.36–2.40) | |
| **Offspring (having one or more children)** | | |
| Yes | 1 | 1 |
| No | 2.96 (1.35–6.47) | 4.05 (1.22–13.33) |

OR: odds ratio; CI: confidence interval; HPV: Human papilloma virus.

aOR: adjusted odds ratio

*Odds ratios adjusted for sex, years of experience and offspring

[a] 6 missing values

[b] 16 missing values

[c] 9 missing values

The vaccines that generate most doubts are those that have been added in the systematic schedule most recently: HPV, introduced in 2008; varicella, in 2016 for infants; and pneumococcus, in 2016 [35]. Other authors have commented that poorly communicated changes in the vaccination schedule and changes in scientific understanding can exacerbate feelings of uncertainty [36]. In addition, there might be other possible factors influencing doubts in these vaccines (e.g. media, religion, or other socio-political factors).

Some authors have seen that changes over time or between regions can motivate HCPs to mistrust in government decisions [36].

Despite having been introduced into the calendar in 2008, the vaccine against HPV generates most controversy and misconceptions, as shown by our results on the perception of its protection and safety. Several studies have reported a low perception of the protection offered by the HPV vaccine by parents, HCPs and girls [10, 22, 23]. This may be related to misinformation in addition to the communication strategies about the infection and the vaccine sometimes portrayed by the media [37]. The low perception of the safety of HPV vaccine among vaccine hesitant nurses, and not shared by the majority of them, is consistent with a study that found that HCPs in PHC not always receive updated and clear evidence based information about the benefits, efficacy and adverse effects of this vaccine, and the authors see a need for strategies to better inform professionals [38].

Another differential factor between hesitant and non-hesitant nurses was the low perception of the severity of immuno-preventable diseases [36]. This could be related to the fact that PHC nurses do not directly treat these diseases as they are attended by paediatricians and, furthermore, serious cases are referred to the hospital. Additionally, these diseases are generally rare [39]. This lack of contact may lead to false beliefs that trivialize vaccination and question the need to vaccinate. Different factors have been suggested to contribute to the increase in mild cases of whooping cough since 2010, including among children who had been correctly vaccinated. For example, the evanescence of the protective effect of the pertussis vaccine or the improvement of epidemiology surveillance could have eroded, among other influencing factors, the perception of effectiveness in vaccines in some nurses [40].

A vast majority of paediatric nurses believed that research advances the improvement of vaccines, that vaccines strengthen the immune system, and that they are one of the safest preventive measures. Even though vaccine-hesitant nurses report mistrust in two issues related to the vaccination calendar: they believed that children received more vaccines than needed and, that some of them are administered too early. These aspects suggest some gaps in the knowledge about the reasons and timing of vaccination. Moreover, we have found an association between the distrust in the pharmaceutical industry, although not with government health authorities, and the nurses' perception about vaccine administration timing. Another study observed that the perceived lack of transparency of administrations could lead to mistrust about changes in the vaccination calendar [10].

Contrary to our assumptions, we observed no differences in knowledge between hesitant and non-hesitant professionals. Both groups showed misconceptions and a moderate proportion of missing values in two questions, which may reflect doubts or information needs. Some European studies highlight the importance of professionals' lack of knowledge about the components of vaccines and the possible consequences of this unawareness [41, 42]. The composition of vaccines is one of the questions most frequently raised by families [27], and it is crucial for paediatric nurses to be familiar with these issues, considering their autonomous-role [43] for families in PHC paediatrics. Spanish and Catalan nurse degree training in vaccination should be reinforced, as we found that less than 50% of nurses reported having enough information about vaccination (data not shown).

Another important factor that could be associated with VH was not having children. The results suggest that nurses that have children were less likely to be vaccine hesitant. A possible interpretation could be that participants that do not have children may assess the risk of these diseases from a hypothetical standpoint, which is reassessed with better information when having children. It would be interesting to study in more depth the relationship between parenthood and a more favourable attitude to vaccination.

Other authors are aware about a less favourable attitude among nurses and towards COVID-19 vaccines.(19) A conducted survey in USA which assess the attitudes towards COVID-19 vaccines among HCPs revealed that the majority of the HCPs choose to wait to

review more data before deciding on personal and only an 8% refuse vaccination [44]. They show differences in vaccine acceptance were observed across demographic lines in their study population, with less acceptance in less served communities. For reasons like these and for many other barriers it is crucial to HCPs to be well informed and thus have the attitudes toward vaccination evidence based. In addition, PHC professionals play a distinction part in educating patients about the vaccine of COVID-19 [45] and their confidence in vaccines plays an important role in their vaccine willingness and the promotion of this to the population now in the pandemic of the COVID-19 [46].

This study has some limitations. Participation was voluntary, which could lead to a selection bias although the percentage of participants was high (>80%). Individuals who did not participate might likely be even more VH. This may modify the estimate effect such a underestimate the magnitude of the problem of the VPH infection [17, 37]. The partial face-to-face administration of the questionnaires could introduce a complacency bias in the responses. A further limitation is that data were collected for a long period of time, 2016–17, so our results on VH, which is known to vary according to place, time, and context [47], may not reflect behaviours before or after that period. Therefore, the results and conclusions of this study may not necessarily apply to other countries or even to other territories in Spain. Even though, some strengths arise, this study accessed to all the paediatric nurses of the PHC centres in Barcelona and allowed to assess the status of a complex multifactorial phenomenon and collected information on the poorly understood issue of VH in paediatric nursing [13]. Our results highlight four factors that may be associated with VH, and used to compare our context with other populations, and that should be addressed.

## Conclusions

Although most paediatric nurses would vaccinate their own children, almost one third display some kind of vaccine hesitancy, mainly related with doubts about HPV and varicella vaccines, as well as some misconceptions. These factors should be addressed to enhance their fundamental role in promoting vaccination among families.

## Supporting information

**S1 Appendix. Original questionnaires information.**
(PDF)

**S2 Appendix. Study questionnaire (Spanish and English version).**
(PDF)

## Acknowledgments

The authors thank the paediatric teams of the public PHC centres of Barcelona who participated in this survey. We also thank Maria Sagué who started this project and participated in the field work.

## Author Contributions

**Conceptualization:** Usue Elizondo-Alzola, Mireia G. Carrasco, Cristina Rius, Elia Diez.

**Data curation:** Camila Andrea Picchio.

**Formal analysis:** Usue Elizondo-Alzola, Elia Diez.

**Investigation:** Camila Andrea Picchio, Cristina Rius.

**Methodology:** Usue Elizondo-Alzola, Mireia G. Carrasco, Cristina Rius, Elia Diez.

**Resources:** Camila Andrea Picchio.

**Supervision:** Mireia G. Carrasco, Laia Pinós, Cristina Rius, Elia Diez.

**Validation:** Mireia G. Carrasco, Elia Diez.

**Visualization:** Usue Elizondo-Alzola, Mireia G. Carrasco, Elia Diez.

**Writing – original draft:** Usue Elizondo-Alzola.

**Writing – review & editing:** Usue Elizondo-Alzola, Mireia G. Carrasco, Laia Pinós, Camila Andrea Picchio, Cristina Rius, Elia Diez.

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
