## [Decision Letter · Decision Letter 0]

8 Mar 2021

PONE-D-20-36118

Vaccine hesitancy among pediatric nurses: Prevalence and associated factors

PLOS ONE

Dear Dr. Elizondo Alzola,

Thank you for submitting your manuscript to PLOS ONE. After careful consideration, we feel that it has merit but does not fully meet PLOS ONE’s publication criteria as it currently stands. Therefore, we invite you to submit a revised version of the manuscript that addresses the points raised below during the review process.

We look forward to receiving your revised manuscript.

Kind regards,

Ray Borrow, Ph.D., FRCPath

Academic Editor

PLOS ONE

Journal Requirements:

2. Please include additional information regarding the survey or questionnaire used in the study and ensure that you have provided sufficient details that others could replicate the analyses.

For instance, if you developed a questionnaire as part of this study and it is not under a copyright more restrictive than CC-BY, please include a copy, in both the original language and English, as Supporting Information. 

If the original language is written in non-Latin characters, for example Amharic, Chinese, or Korean, please use a file format that ensures these characters are visible.

3. Please state whether you validated the questionnaire prior to testing on study participants.

Please provide details regarding the validation group within the methods section.

4. For more information on PLOS ONE's expectations for statistical reporting, please see https://journals.plos.org/plosone/s/submission-guidelines.#loc-statistical-reporting

Please update your Methods and Results sections accordingly.

Reviewers' comments:

Reviewer's Responses to Questions

**Comments to the Author**

1. Is the manuscript technically sound, and do the data support the conclusions?

Reviewer #1: Yes

Reviewer #2: No

2. Has the statistical analysis been performed appropriately and rigorously? 

Reviewer #1: Yes

Reviewer #2: I Don't Know

3. Have the authors made all data underlying the findings in their manuscript fully available?

Reviewer #1: Yes

Reviewer #2: Yes

4. Is the manuscript presented in an intelligible fashion and written in standard English?

Reviewer #1: No

Reviewer #2: Yes

5. Review Comments to the Author

Reviewer #1: Thank you for the invitation to review the following article: Vaccine hesitancy among pediatric nurses: Prevalence and associated factors

General comments:

The data are now 4 years old and in the context of Covid-19 I feel that the introduction and discussion should have additional references that reflect this global pandemic especially in light of the vaccine hesitancy of healthcare professionals

The standard of English is high however there are still some places where it could be improved. I will try to highlight this throughout but think this should be examined also by editorial staff.

Also some spacings should be checked e.g. abstract: Methods . Cross-sectional descriptive study.

I am not sure whether this journal uses OK or US spelling and this again will have to be checked by editors as US spelling seems to be used throughout.

The full questionnaire has not been supplied (to my knowledge) and I feel this should form part of the supplementary material

Abstract:

safety and protection of each vaccine? Safety, and protection offered, of each vaccine

no hesitant…consider replacing with non-hesitant throughout

A 32.1% of them reported vaccine hesitancy? Not clear whether this started off as “Almost a third (32%)….”

especially about the HPV….especially ? regarding? the HPV

low perception….should this be “lower” throughout?

Although most pediatric nurses would vaccinate their own children…. This is not what was measured, it was intention to vaccinate and this should be clear

almost one third display…better to say “reported”

These factors should be addressed to enhance their fundamental….”their” should be explicit

Introduction:

high-income; give example

Line 71-72: VH defined as the reluctance or refusal to vaccinate despite the availability of vaccine, give reference please and comma before rest of the sentence.

Lines 77-78: In Europe, some common reasons against vaccination include the lack of confidence in vaccines, in their administration, in the public health services, and in the pharmaceutical industry. Please correct the sentence as the implication is that “lack of confidence in….” refers to all factors and not just the first one, so insert; as appropriate

Line 79: would suggest consistency; health professionals….later on line 87 becomes healthcare professionals, would suggest using the latter and then abbreviating to HCPs throughout.

Line 82: for example, the FALSE association between the MMR vaccine

Line 90: All vaccines included in the official schedule …….please provide reference

Line 91: so ?effective / positive? communication between families and nurses is crucial

Line 92: Pediatric nurses use an important part of their consultation time to this task, please correct

Lin 99: variables, nurses being more negatively associated with VH, and recommended exploring this…please differentiate here between the statistical term “negatively associated” and the emotional term, that they feel negative about vaccinations as this is not clear

Methods:

Line 110: performed care work…..what does this entail, how is this different to the work of a nurse who does not perform care work?...what are residents? Why were they excluded?

How is temporary nurse defined?

What about part-time nurses?

Line 110: The universe..please use a different term

Line 114: We collected the information using a questionnaire based in literature…upon the literature… please include a copy of the original and the one used in this study in supplementary materials

By whom was this translated? Qualifications?

Line 118: The self-administered questionnaires were made available to nursing staff….where and how and by whom? This is important as you state in the limitations (line 306) “partial face to face administration of the questionnaire”…what does this mean?

Between March 2016 and Feb 2017….how many days, was this a full year? How was the uptake monitored?

Line 121, comma after sex

Line 122; Psychosocial determinants …such as…

Based on theortical models of behaviour….such as….( also here the UK spelling of behaviour is used?)

Were there any questions based upon the level of education of nurses….do all nurses who practice in Spain receive the same training?

Analysis:

Line 156: Student’s t-test, please correct.

We FITTED

Line 160: FEWEST variables

Line 162: consider changing to “no response” to accord with NR

Line 165: please give full reference for Stata software

Results:

Generally would avoid starting a sentence with a number or %

Line 187: H. influenzae b, should be in italics

Line 190: “these” ….which ones, not explicit

Line 207: has “serious” been defined to the participants?

Line 222: we did not

Line 234: again here it is important to define what is meant by “protecting factor” as clearly this is a risk (for public health) that not having children seems to confer to these nurses…consider the use of another term

Discussion:

Line 249-252; is this supposed to be two sentences, please rewrite for clarity.

Line 256: why are you comparing nurses to paediatricians in Switzerland? Especially since you have made the point that paediatricians and nurses in Spain already do not show similarities [15].

Sentence starting…on the other hand…line 256 this is also very unclear as to what point is trying to be made…it would be more useful to use studies that have investigated the same profession…why would you then try to use these data and extrapolate to a completely different group in a different country; line 258: more than 30% of nurses would not recommend vaccines in practice.

Line 262: any reference to support this assumption of suspicion? Or whether communication is suboptimal? And / or whether this is due to the newness or dependent upon completely different factors?

Line 276: please support this with ref

Line 277: please reference this

Line 296: consider rephrasing and using the term “continuing professional development”

Line 298: protective factor…please see earlier comment

Line 205/306: any evidence to support this?

Partial face to face: should be clarified in methods

Overall I think that whilst the emphasis is upon vaccine hesitancy the authors should still mention some findings in terms of vaccine positivity and thus public health

Reviewer #2: This paper addresses an important issue - the view of primary care nurses on vaccinations. However, there are major flaws in the study methodology and the conclusions drawn are not justified by the results. See attached for further detail.

6. PLOS authors have the option to publish the peer review history of their article (what does this mean?). If published, this will include your full peer review and any attached files.

Reviewer #1: No

Reviewer #2: No

---

## [Author Response · Author response to Decision Letter 0]

28 Apr 2021

Dear Editorial team,

Thank you for giving us the opportunity to submit a revised draft of our manuscript titled: ”Vaccine hesitancy among paediatric nurses: Prevalence and associated factors” to Plos One. We really appreciate the time and effort that you and the reviewers have dedicated to providing your valuable feedback on my manuscript. We are grateful to the reviewers for their insightful comments on my paper. We have been able to incorporate changes to reflect most of the suggestions provided by the reviewers. We have highlighted the changes within the manuscript.

Above there is point-by-point the response to the reviewers’ comments and concerns, and answers to the journal requirements questions.

According to Journal Requirements:

Thank you so much for the suggestions and templates recommended. We have checked the Plos One style requirements and make the proper modifications. We have checked our figure through Preflight Analysis and Conversion Engine (PACE) digital diagnostic tool.

 2. Please include additional information regarding the survey or questionnaire used in the study and ensure that you have provided sufficient details that others could replicate the analyses.

For instance, if you developed a questionnaire as part of this study and it is not under a copyright more restrictive than CC-BY, please include a copy, in both the original language and English, as Supporting Information. 

If the original language is written in non-Latin characters, for example Amharic, Chinese, or Korean, please use a file format that ensures these characters are visible.

Thank you for your suggestion. We include now a copy of the questionnaire in Spanish and in English included (S2-Appendix) and more details about the questionnaire and the data collection (above).

 3. Please state whether you validated the questionnaire prior to testing on study participants.

Please provide details regarding the validation group within the methods section.

We appreciate the comment. The questionnaire had not been previously validated, it is on-going. It draws on questions from previously validated studies and are referenced (Salmon et al. 2004, Buxton et al. 2013, Posfay-Barbe et al. 2005). We had specified more detail of the construction and administration of the questionnaire in a previous study we referenced.[15](Picchio CA et al. (2018)). We incorporated the questionnaire from Posfay-Barbe et.al 2005; the reference to the validation article that includes the Buxton et al. 2013 questionnaire, and the reference to the study by Salmon et al. 2004, where the questions of the questionnaire are observed in the tables of the article supplementary data (S1-Appendix). We give more detail of this question above. 

4. For more information on PLOS ONE's expectations for statistical reporting, please see https://journals.plos.org/plosone/s/submission-guidelines.#loc-statistical-reporting

Please update your Methods and Results sections accordingly.

We have updated these sections accordingly to the guideline. Thank you.

Reviewers' comments:

Reviewer's Responses to Questions

 Comments to the Author

1. Is the manuscript technically sound, and do the data support the conclusions?

 Reviewer #1: Yes

Reviewer #2: No

2. Has the statistical analysis been performed appropriately and rigorously? 

Reviewer #1: Yes

Reviewer #2: I Don't Know

3. Have the authors made all data underlying the findings in their manuscript fully available?

Reviewer #1: Yes

Reviewer #2: Yes

4. Is the manuscript presented in an intelligible fashion and written in standard English?

Reviewer #1: No

Reviewer #2: Yes

5. Review Comments to the Author

Reviewer #1: Thank you for the invitation to review the following article: Vaccine hesitancy among pediatric nurses: Prevalence and associated factors

General comments:

1) The data are now 4 years old and in the context of Covid-19 I feel that the introduction and discussion should have additional references that reflect this global pandemic especially in light of the vaccine hesitancy of healthcare professionals

We have contextualised the vaccine hesitancy issue in the pandemic situation of COVID-19, and the relevance rol that healthcare professionals have

Introduction: 

“Given the ongoing COVID-19 pandemic and the need to roll-out mass vaccination campaigns, vaccine acceptance by HCPs is crucial. A dip in vaccine confidence among HCPs could impact implementation for the general population who trust HCPs for their vaccine information and recommendations. In a recent systematic review of COVID-19 vaccine acceptance rates among HCPs (including physicians and nurses), COVID-19 vaccine acceptance rates ranged from 27.7% in the Democratic Republic of the Congo to 78.1% among physicians in Israel, where the rate among nurses surveyed was lower (61.1%).(18) Two studies that dated back to the earlier part of the pandemic (2020 February and March) among nurses in Hong Kong also reported low rates of COVID-19 vaccination acceptance (40.0% and 63.0%).(19)(20) A relevant increase in the acceptance of COVID-19 vaccines among HCP was observed according to a survey; and suggests that the increased intention to receive a vaccine could be influenced by the transparent development process.(21)”

Discussion: 

“Other authors are aware about a less favourable attitude among nurses and towards COVID-19 vaccines (20). A conducted survey in USA which assess the attitudes towards COVID-19 vaccines among HCPs revealed that the majority of the HCPs choose to wait to review more data before deciding on personal and only an 8% refuse vaccination.(44) They show differences in vaccine acceptance were observed across demographic lines in their study population, with less acceptance in less served communities. For reasons like these and for many other barriers it is crucial to HCPs to be well informed and thus have the attitudes toward vaccination evidence-based. In addition, PHC professionals play a distinction part in educating patients about the vaccine of COVID-19 (45) and their confidence in vaccines plays an important role in their vaccine willingness and the promotion of this to the population now in the pandemic of the COVID-19.(46)”

2)The standard of English is high however there are still some places where it could be improved. I will try to highlight this throughout but think this should be examined also by editorial staff.

The mother tongue of one of the authors is english and she has carefully reviewed the paper. 

Also some spacings should be checked e.g. abstract: Methods . Cross-sectional descriptive study.

I am not sure whether this journal uses UK or US spelling and this again will have to be checked by editors as US spelling seems to be used throughout.

Thanks for your comment. We should check the spelling of some words. We have done it now in UK spelling, as far as we know from guidelines both are possible. But we are able to change this according to your preferences and we would be very grateful for comments of improvement in this regard from the journal. Thank you again. 

3)The full questionnaire has not been supplied (to my knowledge) and I feel this should form part of the supplementary material

A copy of the questionnaire in Spanish and in English included now (S2-Appendix) and more details about the questionnaire and the data collection (above).

Now, we answer point-by-point the next sections. We really appreciate all your suggestions and constructive reflexion.

Abstract:

safety and protection of each vaccine? Safety, and protection offered, of each vaccine

no hesitant…consider replacing with non-hesitant throughout

Thanks for the suggestions, we include “offered” to clarify.

“They answered a questionnaire with sociodemographic and behavioral variables: severity and perceived probability of contracting the diseases in the vaccination schedule; safety and protection offered of each vaccine; and beliefs, social norms, and knowledge about vaccines.”

A 32.1% of them reported vaccine hesitancy? Not clear whether this started off as “Almost a third (32%)….”

Thanks for the point. We take out “A” before “32,1%”

especially about the HPV….especially ? regarding? the HPV

Yes, the term “especially” here refers to HPV and varicella vaccines, the two vaccines that reported more VH percentage. To clarify this, we add “nurses”:

“32.1% of the nurses experienced vaccine hesitancy, especially about the HPV (21.9%) and varicella (17.5%) vaccines.”

low perception….should this be “lower” throughout?

We understand that you are referring to line 57. “In the multivariate analysis, hesitancy was associated with low perception of the severity of whooping cough…”

If this is what you are referring to, we think that this should be “low” and not “lower” since this is how we have categorized perception (high and low). 

Although most pediatric nurses would vaccinate their own children…. This is not what was measured, it was intention to vaccinate and this should be clear

almost one third display…better to say “reported”

Thank you for your comment, we modify “display” for “reported” and clarify that we asked for the intention of vaccination.

“Although most pediatric nurses had the intention to vaccinate their own children, almost one-third reported some kind of vaccine hesitancy, mainly related to doubts about HPV and varicella vaccines…”

These factors should be addressed to enhance their fundamental….”their” should be explicit

We agree with the necessity of clarifying who we are referring to using “their”.

“These factors should be addressed to enhance nurses’ fundamental role in promoting vaccination among families.”

Introduction:

high-income; give example

We appreciate the clarification proposed. We were referring to the immune-preventable diseases just indicated. We better clarify as you suggest. 

“However, in several high-income European countries, such as Italy and France, immunization rates of some immuno-preventable diseases such as measles have declined in recent decades, ...”

Line 71-72: VH defined as the reluctance or refusal to vaccinate despite the availability of vaccine, give reference please and comma before rest of the sentence.

Thank you, we add this comma: “VH defined as the reluctance or refusal to vaccinate despite the availability of vaccines, threatens to reverse progress made in tackling immuno-preventable diseases”.

Lines 77-78: In Europe, some common reasons against vaccination include the lack of confidence in vaccines, in their administration, in the public health services, and in the pharmaceutical industry. Please correct the sentence as the implication is that “lack of confidence in….” refers to all factors and not just the first one, so insert; as appropriate

Thank you, we add this comma: “In Europe, some common reasons against vaccination include the lack of confidence in vaccines; in their administration, in the public health services, and in the pharmaceutical industry.”

Line 79: would suggest consistency; health professionals….later on line 87 becomes healthcare professionals, would suggest using the latter and then abbreviating to HCPs throughout.

We appreciate your comment, and we totally agree with the need of consistency for the term “healthcare professionals”. We have agreed to incorporate the abbreviation HCPs

“In the last decade, social networks and some digital media have contributed to expanding these doubts and to eroding families' trust in healthcare professionals (HCPs).”

Line 82: for example, the FALSE association between the MMR vaccine

Thank you for the suggestions. Totally agree and need to explain an erroneous belief in our context as well as in others.

“In France, 36% of parents question the safety of vaccines [9], and, in Spain, controversies are quite similar to those in other countries: for example, the false association between the MMR vaccine and autism is not uncommon,...”

Line 90: All vaccines included in the official schedule …….please provide reference

Line 91: so?effective / positive? communication between families and nurses is crucial

Thanks for these two comments. We had added the reference later in the text, but it must be added here. It is also necessary to add an adjective to explain how the communication should improve the vaccination coverage. 

“All vaccines included in the official schedule are recommended, not required, so positive communication between families and nurses is crucial to maintain high vaccination coverages. [15]”

Line 92: Pediatric nurses use an important part of their consultation time to this task, please correct

“Pediatric nurses use an important part of their consultation time for tasks related to vaccination, …”

Line 99: variables, nurses being more negatively associated with VH, and recommended exploring this…please differentiate here between the statistical term “negatively associated” and the emotional term, that they feel negative about vaccinations as this is not clear

We appreciate this good point highlighted here. 

“The study found some differences between paediatricians and paediatric nurses and nurses reported more VH (adjusted odds ratio: 2.0; 95% CI: 1.1–3.7), and recommended exploring this phenomenon in greater detail.(17)”

Methods:

Line 110: performed care work…..what does this entail, how is this different to the work of a nurse who does not perform care work?...what are residents? Why were they excluded?

How is temporary nurse defined?

What about part-time nurses?

We were referring to the care work with patients and not those healthcare professionals that work in management. We should include “with patients” for clarification, and add that those working in management were excluded. On the other hand, residents are those in a training period of a speciality, that in the case of Spain nurses do it for a period of two years in pediatrics, mental health, community health, matron nurses…We decide to exclude those because they are in a training process as students. Finally, temporary nurses were those who work in the primary healthcare center with a short hiring period. However, those with a part-time working day were included, because they work continuously in the center and they are not neither of the other options. 

“We included nurses who performed care with patients (not nurses working in management) and excluded students, residents (because they are in a training period), and temporary nurses (because of their short working periods in the centre).”

Line 110: The universe..please use a different term

We change it for the term “population sample”: 

“ The population sample was 165 nurses.”

Line 114: We collected the information using a questionnaire based in literature…upon the literature… please include a copy of the original and the one used in this study in supplementary materials

By whom was this translated? Qualifications?

We include now in supplementary materials a copy of the questionnaire used in the study in Spanish and another in English (S2-Appendix). According to the suggestion to include a copy of the original questionnaire, we incorporated the questionnaire from Posfay-Barbe et.al 2005; the reference to the validation article that includes the Buxton et. 2013 questionnaire; and the reference to the study by Salmon et.al. 2004, where the questions of the questionnaire are observed in the tables of the article in supplementary data (S1-Appendix).

The questionnaire had not been previously validated, it is on-going. It draws on questions from previously validated studies and are referenced (Salmon et.al. 2004, Buxtonet.al 2013, Posfay-Barbe et.al 2005).

The questionnaire was translated by the research team. The questionnaire was translated into Spanish and Catalan by the research team and culturally adapted using the cognitive debriefing method (reference include Nixon-2015). 

We had specified more detail of the construction and administration of the questionnaire in a previous study we referenced (Picchio CA et al. 2018). And, also now, we include more detail in our manuscript (above, after line 118 you can see how all this date has been integrated). 

Line 118: The self-administered questionnaires were made available to nursing staff….where and how and by whom? This is important as you state in the limitations (line 306) “partial face to face administration of the questionnaire”…what does this mean?

Between March 2016 and Feb 2017….how many days, was this a full year? How was the uptake monitored?

We recognized that in the manuscript there was not enough information according to the data collection. We decided to refer to this information to the previous study of Camila Picchio et al. 2019. However, now we agree with you and the necessity to clarify all the points you have suggested here. 

The questionnaires were self-administered by pediatric nurses in the primary care center in a date and time previously agreed between researchers and a contact person from each centre. Researchers (usually two) went to the primary care centre on the date agreed to administer the survey on paper and in-person (answer refers to the comment in line 306 about limitations about partial face to face). Researchers explained the objective of the study, provided informed consent forms, and finally distributed surveys which were available in Spanish and Catalan to every health care professional present. Researchers waited in a corner until everyone completed the survey and did not engage in conversation with study participants so as to not influence responses. Data was entered using TeleForm® software. Any errors detected were compared with the original questionnaires and changed. The questionnaire data was imported to Stata by one researcher. This person was responsible for cleaning the data in this software.

“Data Collection

We collected the information using a questionnaire based in literature,[6,18,19] (S1-Appendix) translated into Catalan and Spanish and culturally adapted using the cognitive debriefing method by the research team.[20] Cognitive debriefing is a process where representatives of the target population actively test the translated questionnaires to determine whether respondents would understand the questionnaire as easily as the primary version would be understood.[21] The self-administered questionnaires were made available to nursing staff between March 2016 and February 2017. The questionnaires were given on paper to the participating nurses. After an initial email sent out to all PHC centres, an 11-month period was needed to schedule the visits on days and times suitable for all. Additionally, PHC centres who did not initially respond were followed-up up to 6 times via email in order to include them in the study. In reaching out to each centre referent and speaking with them on a personal level, researchers were able to contact and reach almost all PHC centres (only one was excluded) with paediatric departments in the public health system in the city of Barcelona. Two researchers attended the PHC centre and after a brief explanation of the study objectives distributed the questionnaire for participants to respond on their own. Finally, the participants answered the questionnaires and had however much time they needed to complete them, but most completed it in about 15 minutes. Data was entered using TeleForm® software. Any errors detected were compared with the original survey and changed. This data was imported to Stata version 12.0 (Stata Corp. 2011. Stata Statistical Software: Release 12. College Station, TX: StataCorp LP) by one researcher who cleaned the data in this software.” 

Line 121, comma after sex

Thanks for point out this mistake: “ (age, sex, years of experience)”

Line 122; Psychosocial determinants …such as…

Based on theoretical models of behaviour….such as….( also here the UK spelling of behaviour is used?)

Were there any questions based upon the level of education of nurses….do all nurses who practice in Spain receive the same training?

We appreciate all these comments. Thank you for the spelling suggestion. We adjust it to UK spelling and do it consistently in all the manuscripts. We are available to listen to any suggestions regarding the language.

According to the training in Spain, nurses received practically the same training during the degree. Then, some specialize in pediatric nursing, so they gain more specific knowledge. And then the continuing education is always offered by the healthcare system, but also depends on each nurse's interest. We do not ask in the questionnaire about the grade of education in terms of pediatric specialty or course. However, we included two extra items related to training: 1) if they had enough training and materials to appropriately doubt about vaccination and, 2) which type of materials would prefer to resolve doubts to parents (print material, online material, face to face sessions). Therefore, the 50% of nurses that reported having enough information about vaccination. We add a clarification in the manuscript, we hope it will be useful. 

“Spanish and Catalan nurse degree training in vaccination should be reinforced, as we found that less than 50% of nurses reported having enough information about vaccination (data not shown).” 

Analysis:

Line 156: Student’s t-test, please correct.

We FITTED

Line 160: FEWEST variables

Line 162: consider changing to “no response” to accord with NR

Thank you so much for all these spelling and grammatical errors pointed (for line 156, 160 and 162) that improve the understanding and coherence. We have all included. 

Line 165: please give full reference for Stata software

We add in the manuscript the Stata reference used for the analysis: StataCorp. 2013. Stata Statistical Software: Release 13. College Station, TX: StataCorp LP

Results:

Generally would avoid starting a sentence with a number or %

We appreciate this recommendation, and applied in the manuscript. We make some improvements in line 171 and 178:

Line 171: “83% of the 165 nurses participated in the study” change to 

“Of the total number of nurses working in the PHC of the public health system (n=165), 83% (n=137) participated in the study. Of these, 96.4% of them were women, with a mean age 47.7 years (SD: 10.2), 72.3% had children, and had an average of 23.8 years (SD: 10.5) of professional experience (Table 1).”

Line 178: “Of the nursing staff 67.9% stated their total agreement to vaccinate their children with all the vaccines in the systematic schedule. Some (32.1%) reported having doubts, delaying the administration of certain vaccines or avoiding vaccinating their children with at least one of the vaccines in the systematic schedule.”

Line 187: H. influenzae b, should be in italics

Thank you for this point, H. influenzae b corrected now.

Line 190: “these” ….which ones, not explicit

Thank you. In line 190 we refer to all vaccines considered in this study, which are all the vaccines of the systematic vaccines schedule in Catalonia. We clarify with reference to “all” vaccines.

“More than 90% of vaccine-hesitant nurses perceived all these vaccines to be very safe, except for the HPV vaccine,…”

Line 207: has “serious” been defined to the participants?

No, it was not defined this term to participants. It was left to the interpretation of each participant because it is asking about a belief and also all were under the same healthcare professional context and a common basic training.

Line 222: we did not

Thank you for the correction.

Line 234: again here it is important to define what is meant by “protecting factor” as clearly this is a risk (for public health) that not having children seems to confer to these nurses…consider the use of another term

Thanks for questioning the meaning. The results show that not having children could be an associated factor of vaccine hesitancy. On the other hand, having children could be a protection against vaccine hesitancy. We explained the result of not having children erroneously or could become confusing since the protection factor is “having children” and not “no having children”. Therefore, we suggest an improvement for this explanation in line 234. We suggest: 

“Nurses that did not have children were more likely to report hesitancy (aOR: 4.05; 95% CI:1.22-13.3).”

Discussion:

Line 249-252; is this supposed to be two sentences, please rewrite for clarity.

Thank you. 

“The vaccines that generated most doubts were those against HPV, varicella, pneumococcus and hepatitis A. Vaccine-hesitant nurses had a lower perception of the risk for some diseases, a lower perception of the benefit of the varicella and HPV vaccines, and generally more unfavorable beliefs about vaccination (e.g. the time of administration or the number of vaccines) than non-hesitant nurses.”

Line 256: why are you comparing nurses to paediatricians in Switzerland? Especially since you have made the point that paediatricians and nurses in Spain already do not show similarities [15].

Sentence starting…on the other hand…line 256 this is also very unclear as to what point is trying to be made…it would be more useful to use studies that have investigated the same profession…why would you then try to use these data and extrapolate to a completely different group in a different country; line 258: more than 30% of nurses would not recommend vaccines in practice.

Referring to studies of nurses would be ideal, both in our countries and in others. However, during the study and during the time of writing this manuscript, we did not find anything more comparable to our target population. Now, again, we have updated a search and, as far as we have found, there are still missing studies published targeting nurses involved in the pediatric population. Therefore, healthcare professionals in primary care are the most similar population comparable since they share common aims in the care of patients in the community. Obviously, we are aware of the limitations this has. Moreover, although they have different roles in Spain, as in other countries, these healthcare professionals are complementary. Therefore, we are still considering the possibility of comparing our results with the ones that other European countries have published in general practitioners or pediatricians. We would like to add that this means the lack of knowledge and the need to research about pediatric nursing perception in this area. In addition, we appreciate your comment because we think it is necessary to contextualize these lines and explain to the reader why we are comparing to that. Thus, thank you again.

According to the comment of line 256, on the assumption that Barcelona’s pediatric nurses would have a similar behavior, and more than 30% of nurses would not recommend vaccines in practice, we have thought about the subject and it would be better not to extrapolate the data as you recommend. We eliminate that sentence. Thank you.

“Although we would like to compare our results with studies with the same aim targeting, to our knowledge, there are no other published studies with these characteristics, highlighting the research gap in studying VH in this population. We therefore put our findings in the context of studies targeting HCPs (in general) working in PHC. Compared with other European countries, Barcelona paediatric nurses appear to be less likely to have the intention to vaccinate their offspring according to the systematic childhood vaccination schedule than professionals in countries like Switzerland (22), where 95% of paediatricians would vaccinate. On the other hand, general practitioners in France are less likely to recommend vaccines to their patients than to their offspring (32). Other authors, also in France, found that HCPs have divergent immunization attitudes toward their relatives and their patients when asking about the intention of vaccinating their own children(33), especially when considering the newest and most controversial vaccines, like the HPV vaccine. A cross-sectional study done in Croatia through a self-administered questionnaire on attitudes, beliefs and behaviours relating to vaccination among HCPs, including paediatric nurses, reported that nurses were more likely than paediatricians to be vaccine-hesitant (aOR = 5.73, 95%CI = 2.48–13.24). Therefore, in general, our results are similar to those reported across other European studies, but it seems to be slightly higher VH in nursing than in paediatricians.[34]”

Line 262: any reference to support this assumption of suspicion? Or whether communication is suboptimal? And / or whether this is due to the newness or dependent upon completely different factors?

Thank you for your questions and comments. We agree with the necessity of improving the understandability of what we want to say. To give more weight to the possible influencing factor of communication we add other recent references, even though some previous such as Larson et al. 2011. Therefore, we add some lines to address this.

“The vaccines that generate most doubts are those that have been added in the systematic schedule most recently: HPV, introduced in 2008; varicella, in 2016 for infants; and pneumococcus, in 2016.[35] Other authors have commented that poorly communicated changes in the vaccination schedule and changes in scientific understanding can exacerbate feelings of uncertainty [36]. In addition, there might be other possible factors influencing doubts in these vaccines (e.g. media, religion, or other socio-political factors). Some authors have seen that changes over time or between regions can motivate HCPs to mistrust in government decisions [36].” 

Line 276: please support this with ref

“Another differential factor between hesitant and non-hesitant was the low perception of the severity of immuno-preventable diseases (Yaqub et.al 2014).”

In the article of Yaqub et.al 2014:

“Lack of awareness, low perceived severity of illness and a belief in alternative medicine were often cited as reasons for hesitancy.”

Yaqub O, Castle-Clarke S, Sevdalis N, Chataway J. Attitudes to vaccination: A critical review. Soc Sci Med. 2014;112(1):11.

Line 277: please reference this

“This could be related to the fact that primary care nurses do not directly treat these diseases as they are attended by pediatricians and serious cases are referred to the hospital, or, importantly, because these diseases are generally rare (Dubè et al. 2013).”

In the article of Dubé et al. 2013:

“Because vaccination programs have been successful, VPD are becoming less visible and many individuals, as well as health professionals, have no first-hand knowledge of the risks of the diseases.”

Eve Dubé, Caroline Laberge, Maryse Guay, Paul Bramadat, Réal Roy & Julie A. Bettinger (2013) Vaccine hesitancy, Human Vaccines & Immunotherapeutics, 9:8, 1763-1773, DOI: 10.4161/hv.24657

Line 296: consider rephrasing and using the term “continuing professional development”

Thank you so much for the comment:

Finally, we have decided to remove the sentences: “Besides, the continuing nursing professional development in Spain is optional.”

Line 298: protective factor…please see earlier comment.

We appreciate the suggestion. We add an explanation of the protective factor in results. And we add some detail in these lines: 

“Another important factor that could be associated with VH was not having children. The results suggest that nurses that have children were less likely to be vaccine hesitant. A possible interpretation could be that participants that do not have children may assess the risk of these diseases from a hypothetical standpoint, which is reassessed with better information when having children. It would be interesting to study in more depth the relationship between parenthood and a more favourable attitude to vaccination.”

Line 205/306: any evidence to support this?

Partial face to face: should be clarified in methods

We agree with the need of clarifying the term partial face to face in a better way. We have now included a better description of the questionnaire, so the reader could understand the dissemination of the questionnaire among nurses. Therefore, the sentence in lines 306 may be understandable now. 

“The partial face-to-face administration of the questionnaires, could introduce a complacency bias in the responses, ”

Overall I think that whilst the emphasis is upon vaccine hesitancy the authors should still mention some findings in terms of vaccine positivity and thus public health

Thank you for your suggestion. We appreciate all your suggestions and we have highlighted the findings in terms of vaccine positivity and as a public health issue. 

Reviewer #2: This paper addresses an important issue - the view of primary care nurses on vaccinations. However, there are major flaws in the study methodology and the conclusions drawn are not justified by the results. See attached for further detail.

Reviewers' comments:

Abstract:

Line 45-46: Objective: The associated factors influencing vaccine hesitancy are multiple and complex and are not all captured by this survey. Rephrase the objective to remove this part. Is the objective a prevalence of vaccine hesitancy and views on certain vaccines?

We appreciate this constructive reflexion. Our objective is to describe the prevalence of vaccine hesitancy and some related psychosocial determinants of the systematic vaccination schedule among nurses. The questionnaire included Health Belief Model) related with participants' perception of the severity and probability of contracting each immuno-preventable disease, and the safety and protection conferred for each Ag in the schedule). We detail this in methods clearly and we proposed an improve in these two senses. 

“Objective. This study describes the prevalence of vaccine hesitancy and some related psychosocial determinants of the Spanish recommended vaccinations among pediatric primary care nurses in Barcelona.”

Line 51: dichotomized into not hesitant

Thanks for indicating the mistake in typing “not”.

Abstract: Results: 

Line 55: Remove A from before 32.1%. 

Does sentence refer to the nurses themselves being hesitant about HPV and varicella, should state so more clearly than ‘reported’ which could mean they report patient/public hesitancy?

Thank you for your suggestion. We agree with your comment and we add for more clarity:

“32.1% of the nurses experienced vaccine hesitancy, especially about the HPV (21.9%) and varicella (17.5%) vaccines.”

Line 57-59: Brackets in wrong place around the confidence intervals.

[aOR: 3.88; 95%CI:1.32-11.4]

The statements supported by the Odds Ratios are perhaps too strong in their conclusion as the confidence intervals are very wide. I would recommend adjusting the strength of statement to be more suggestive, or could possibly be linked. I would not draw a strong association from these OR and CI. 

Thank you for your comment, which we have considered very valuable and therefore we try to adjust the strength of the statements:

“The multivariate analysis suggests associations between hesitancy and low perception of the severity of whooping cough [aOR: 3.88; 95%CI:1.32-11.4], low perception of safety of the HPV vaccine [aOR:8.5;95%CI:1.24-57.8], the belief that vaccines are administered too early [aOR:6.09;95%CI:1.98-18.8], and not having children [aOR:4.05;95%CI:1.22-13.3].

The Abstract does not refer to findings where there was a large degree of agreement, such as results in lines 218-220, and instead has highlighted certain ‘controversial’ findings which I believe is misleading. Large agreement on vaccine effectiveness and safety was reported in the results which is important. A balanced view of the findings is not present in the current abstract. 

We agree with the usefulness of adding positive agreements of the findings, and the need to show a balance of them. We really appreciate this view. Therefore, we proposed to include in the results.

“Although most paediatric nurses had the intention to vaccinate their own children, almost one-third reported some kind of vaccine hesitancy, mainly related to doubts about HPV and varicella vaccines, as well as some misconceptions. These factors should be addressed to enhance nurses’ fundamental role in promoting vaccination to families.” 

Line 61: related to

Thanks for indicating the mistake in the preposition. 

Line 70: what disease is referred to here? Provide some examples of countries and immune-preventable diseases to explain. 

We appreciate the clarification proposed. We were referring to the immune-preventable diseases just indicated. As it has been suggested, we better clarify:

“However, in several high-income European countries, such as Italy and France, immunization rates of some immuno-preventable diseases such as measles have declined in recent decades, which has contributed to recent outbreaks of the measles disease[2].

Line 69 and 73: immune-preventable or vaccine-preventable diseases – pick one for consistent use throughout paper. 

We agree with your comment on the need of consistency in the use of terms. We just pick immune-preventable.

Line 82-83: rephrase the following “association between the MMR vaccine and autism is not uncommon”. This is a misconception and should be clearly presented as such, in its current phrasing it suggests there is a link between MMR and autism and is inaccurate. 

In this example, we try to present an erroneous belief in our context as well as in others. We agree that we should present this as a false belief. Therefore, we propose: 

“for example, the false association between the MMR vaccine and autism is not uncommon, 

Line 96-100: Could the authors expand here on what the key findings of this study were? 

Thank you so much for pointing out the need for a wider explanation of these findings. 

“A study among pediatric PHC teams in Barcelona found that 25% of the HCPs involved in vaccination had doubts about some of the vaccines on the systematic vaccination schedule, especially the human papillomavirus vaccine The authors found that up to 40% of the respondents lacked specific knowledge about vaccination. The study found some differences between pediatricians and pediatric nurses; nurses reported more VH ((aOR)=2.0; 95% c(95% CI):1.1–3.7), and recommended exploring this phenomenon in greater detail.[17]”

Line 102-103: as per comment on the paper objective I don’t believe this survey captures information on factors associated with VH in the study population. 

Following the recommendations for improvement, we specify that the factors captured in this study are factors related to psychosocial determinants based on theoretical models of behaviour.

“The aim of the current study was to determine the prevalence of VH in regards to the vaccines in the systematic childhood vaccination calendar of Catalonia (Spain) and to explore some of its psychosocial determinants among PHC paediatric nurses in the city of Barcelona.” 

Line 110: replace universe with population sample. 

The population sample was 165 nurses.

Line 107-111: Not enough detail provided on the study population. How did the researchers access their contact details or disseminate information? Were they registered nurses with a national nursing authority or board?

Explain cross-sectional study, should outline more information that it is a survey from the outset of methods. 

Thank you so much for the suggestion. We thought that this information was covered with previous studies. However, we also see the benefits of adding here more detail about the study population. Therefore, we add: 

“We conducted a cross-sectional study. The study population included all the paediatric nurses working in the 41 public PHC centres with paediatric departments in the city of Barcelona in 2017. We included nurses who performed care with patients (not nurses working in management) and excluded students, residents (because they are in a training period), and temporary nurses (because of their short working periods in the centre). The population sample was 165 nurses.”

Data collection Lines 114-119:

1) Insufficient information on the development of the questionnaire. Was the questionnaire in reference 15 validated, previously disseminated? For what population? How did the researchers decide on the relevance of all the questions, were any questions removed from the original questionnaire? 

The questionnaire was addressed to pediatric healthcare professionals (pediatricians and pediatric nurses) who work in the public healthcare system and are directly involved with systematic childhood vaccination. The questionnaire had not been previously validated, it is on-going. It draws on questions from previously validated studies and are referenced (Salmon et al. 2004, Buxton et al. 2013, Posfay-Barbe et al. 2005). We had specified more detail of the construction and administration of the questionnaire in a previous study we referenced.[17](Picchio CA et al. (2018)). We incorporated the questionnaire from Posfay-Barbe et.al 2005; the reference to the validation article that includes the Buxton et al. 2013 questionnaire; and the reference to the study by Salmon et al. 2004, where the questions of the questionnaire are observed in the tables of the article supplementary data (S1-Appendix).

2) Who translated the questionnaire? How was this tested for clarity? 

The questionnaire was translated into Spanish and Catalan by the research team and culturally adapted using the cognitive debriefing method (reference include Nixon-2015). Cognitive debriefing is a process where representatives of the target population actively test the translated questionnaires and allow researchers to determine whether respondents would understand the questionnaire as easily as the English version would be understood (Language Scientific. Cognitive debriefing explained. Medford: Language Scientific. Available from: http://www.languagescientific.com/cognitive-debriefing-explained/) 

3) How were the questionnaires sent to nurses, by paper, online, by email or phone message? 

The questionnaires were self-administered by pediatric nurses in the primary care center in a date and time previously agreed between researchers and a contact person from each centre. Researchers (usually two) went to the primary care centre on the date agreed to administer the survey on paper and in-person. Researchers explained the objective of the study, provided informed consent forms, and finally distributed surveys which were available in Spanish and Catalan to every health care professional present. Researchers waited in a corner until everyone completed the survey and did not engage in conversation with study participants so as to not influence responses. Data was entered using TeleForm® software. Any errors detected were compared with the original questionnaires and changed. This data was imported to STATA and cleaned. 

4) Were they sent out at one point in time or on different days in the primary care centres? Explain why the survey was sent out over a 11 month period, why was this done, surely there would be different views over this length of time potentially due to media/social influences over the course of this year that can influence people's views? A questionnaire should capture all respondents’ views at the same point in time. This is a major flaw. 

“The collection of information was adapted to the availability of the centers. After an initial email sent out to all PHC, a 11-month period was needed to schedule the visits on days and times suitable for all. Additionally, PHC centers who did not initially respond were followed-up up to 6 times via email in order to include them in the study. In reaching out to each centre referent and speaking with them on a personal level, researchers were able to contact and reach almost all PHC centers (only one was excluded) with pediatric departments in the public health system in the city of Barcelona.”

5) Was the questionnaire collected on paper and if so who input the data into Stata, was it double checked and data cleaned?

The questionnaire data was imported to Stata by one researcher. This person was responsible for cleaning the data in this software.

All this suggestions have been included in the manuscript in the data collection section as: 

“Data Collection

We collected the information using a questionnaire based in literature,[6,18,19] (S1-Appendix) translated into Catalan and Spanish and culturally adapted using the cognitive debriefing method by the research team.[20] Cognitive debriefing is a process where representatives of the target population actively test the translated questionnaires to determine whether respondents would understand the questionnaire as easily as the primary version would be understood.[21] The self-administered questionnaires were made available to nursing staff between March 2016 and February 2017. The questionnaires were given on paper to the participating nurses. After an initial email sent out to all PHC centres, a 11-month period was needed to schedule the visits on days and times suitable for all. Additionally, PHC centres who did not initially respond were followed-up up to 6 times via email in order to include them in the study. In reaching out to each centre referent and speaking with them on a personal level, researchers were able to contact and reach almost all PHC centres (only one was excluded) with paediatric departments in the public health system in the city of Barcelona. Two researchers attended the PHC centre and after a brief explanation of the study objectives distributed the questionnaire for participants to respond on their own. Finally, the participants answered the questionnaires and had however much time they needed to complete them, but most completed it in about 15 minutes. Data was entered using TeleForm® software. Any errors detected were compared with the original survey and changed. This data was imported to Stata version 12.0 (Stata Corp. 2011. Stata Statistical Software: Release 12. College Station, TX: StataCorp LP) by one researcher who cleaned the data in this software.” 

Line 124: vaccinable diseases….see comment regarding lines 69 and 73.

We appreciate the correction in these terms. We decide to use “immune-preventable diseases” instead of “vaccinable diseases.”

Line 103 and line 125: What was the geographical area of work of the study nurses included in the study? Barcelona is one area within a larger Catalonia, were the nurses working all over Catalonia or not? 

On the one hand, we would like to clarify that the Catalonia vaccine schedule applied to all catalan regions and their cities, including Barcelona. Therefore, in line 125 we include a clarification in this sense: 

“vaccines on the systematic schedule in Catalonia[17], which are applicable throughout the region including the city of Barcelona"

On the other hand, the area of work of the study nurses was the city of Barcelona. We specify this in line 103: 

“The aim of the current study was to determine the prevalence of VH about the recommended vaccines in Spain and to study some of its psychosocial determinants among PHC pediatric nurses in the city of Barcelona (Spain).”

Line 130 – 135: Why do the authors talk about antigens? Would the term vaccine be more clear and consistent? What was included in the questionnaire?

In the questionnaire we use the term vaccine because of its ease of understanding. However, in manuscript we use the term antigen referring to the vaccine because vaccines have antigens as their active principle but some of them have several antigens (in combined vaccines). Considering the population to which this manuscript is addressed we considered it more adequate. Even though, we will accept your suggestions and use the word “vaccine” for consistency and clarity

“…vaccinate their offspring for each of the 14 vaccines on the vaccination calendar.” (line 130)

“…the answer was "no", "I have doubts" or "I would do it later" for one or more vaccines , ...” (line 134)

Line 135: clarify exactly what responded otherwise means.

If the answer was “yes” to the question "If you had a child today, would you agree for them to receive the vaccines on the current systematic schedule?" we classified it as “non-hesitancy”. Thus, we clarify what we want to express with “otherwise: 

“and “non-hesitancy” if they responded yes to all vaccines, what means they agree for their child to receive the vaccines.”

Several health behaviour theories are being mentioned in the methods e.g. line 131 Theory of Reasoned Action, Line 137 HBM, with no justification of why they are being applied. 

Was the questionnaire not based on a previous questionnaire (ref 15) and did this questionnaire refer to these theories or not in it’s development?

The study on which we based our questionnaire used the variables susceptibility, severity efficacy, and safety. These constructs are based in the Health Belief Model [22], which refers to subjective assessment of a certain expectation about a health behaviour that people have disposition to adopt. We also used the intention to vaccinate, a good predictor of a behaviour, which is a construct of the

Theory of Reasoned Action and the Integrated Behavioral Model.[21] Based on these models, the study variables included the participants’ perception of the severity and probability of contracting each immuno-preventable disease and the safety and protection conferred by the vaccines.

The section on Variables is confusing and difficult to interpret without a copy of the questionnaire to outline what was on the final questionnaire? Again, it is not clear how the several health theories can all be applied in one questionnaire. No explanation of the theory, relevance or what elements of the theory are used to guide the questions. 

Thanks for your comments. We add the questionnaire as supplement data.

In relation to the theories, we explicit which theories are used for each question: the intention to vaccinate is based on the Theory of Reasoned Action which is a theory of planned behavior; the probability of contracting the disease, severity of the illness, safety of the vaccine, and protection conferred by the vaccine are based on the Health Belief Model. 

Line 154-160: what variables were used in the logistic regression model? 

In the logistic regression models we used all the variables statistically significant in the bivariate analysis,as described in tables 2 and 3. In line 158 we comment this information. 

“We fitted logistic regression models using the variables statistically significant in the bivariate analysis and adjusted for sex, years of experience and offspring. The variables included in the models are described in tables 2 and 3.” 

Line 169: provide actual number of 83% of 165, how many responses?

Thanks for your comment. 137 nurses participated out of 165 (83% of participation). The number 137 is provided in table 1, 2, and 3. We add this number in the sentence: 

“Of the total number of nurses working in the PHC of the public health system (n=165), 83% (n=137) participated in the study. Of these, 96.4% of them were women, with a mean age 47.7 years (SD: 10.2), 72.3% had children, and had an average of 23.8 years (SD: 10.5) of professional experience (Table 1).”

Line 170: remove ‘was’ from end of sentence. 

Thanks for pointing out this misprint.

Table 1: Does Vaccine hesitancy categorisation here mean they were hesitant of all vaccines or at least vaccine? Did the questionnaire capture that level of detail as suggested by lines 176-179.

As we mention in methods, vaccine hesitancy was defined as “yes” when the participant answered “no”, "I have doubts" or "I would do it later" for one or more vaccines to the question "If you had a child today, would you agree for them to receive the vaccines on the current systematic schedule?". This means that if at least one of the 14 vaccine answers were “no", "I have doubts" or "I would do it later” that participant was categorized as “vaccine hesitancy”. As we listed each vaccine, we know which vaccine generated the greatest hesitancy. Reading your question, we propose a clarification in table 1, in the footnote according to the “yes” and “no” categorization of the “vaccine hesitancy” variable:

1 Yes: Participants responded “no" or "I have doubts" or "I would do it later” for at least one vaccine of the 14 listed vaccines.

2 No: Participants responded “yes” to all vaccines.

Line 185: what are the 14 diseases in the schedule?

It refers to the 14 immuno-preventable diseases covered by the 14 vaccines of the systematic schedule in force in Catalonia. Regarding your question, we think that would be clearer for the reader the following sentence:

“In 7 of the 14 diseases of the vaccination schedule the perception of the risk of infection was similar in non-hesitant nurses and hesitant nurses. Hesitant nurses had a low perception risk of infection for 5 for diphtheria, whooping cough, polio, measles and HPV. They also reported a low perception of severity for 7 of them: whooping cough, H. influenzae b, meningococcal disease, hepatitis A, measles, mumps, and varicella (Table 2).”

Line 190: what does these vaccines refer to? All 14 vaccines?

It refers to all vaccines considered in this study. Your suggestion improves much more the understanding of the sentence and what we refer to. Therefore, we add “all” to the sentence. 

“More than 90% of vaccine-hesitant nurses perceived all these vaccines to be very safe, except for the HPV vaccine,…”

Line 191 and 194: provide actual numbers in brackets with 76.5%, 97.7% and 90%

We appreciate your suggestion, and we consider it interesting for the reader to have this information.

“which was considered to be very safe by 76.5% (n=26) of the vaccine-hesitant nurses, and by 97.7% (n=85) of the non-hesitant nurses (p<0.001)”

Table 2: title should read that it is nurses views of these factors/ agreement with these statements. 

Thank you so much for your suggestion.

Table 2. Nurses’ views on disease susceptibility and severity, and vaccine safety and protection by hesitancy Barcelona 2016-17 (N=137)

Table 2, footnote e: was the question addressed to the parent of an 8 year old child? How were these questions phrased? Were the nurses asked their own opinion of HPV susceptibility, severity, safety and protection, not in the context of considering their own child? This is not clear. 

Thank you for the comment. The footnote does not clearly specify why there is no answer in this subdimension of severity of the diseases. 

In response to the question, we asked in the questionnaire: “This question is about the severity of vaccinable diseases. How serious do you consider it is that an 8-year-old non-immunized child catches the following diseases?''. The question addressed pediatric nurses (never to parents). The questionnaire avoided asking about the human papillomavirus vaccine severity perception because HPV does not cause an acute disease as in all the other immuno-preventable diseases asked.

We proposed for these questions the age of 8 years because they should have already received at least two doses of the vaccines included in the schedule, except for HPV which is given later.

We consider adding more detailed information to clarify this issue.

“e HPV severity was not considered given that the question in this block referred to an 8 year old child and it is understood that at this age they do not get the virus, because HPV that not cause an acute disease”

We are available to try to clarify more if needed. 

Line 206: hesitant professionals? Nurses were the only group in the sample – replace. 

We appreciate the correction of this detail.

“Hesitant nurses”

Lines 222-228: was there a ‘do not know’ option answer here? Did respondents who did not enter a response here, respond to other questions in this section or not? It cannot be suggested that missing values mean the respondent did not know the answer, therefore this paragraph does not present ‘notable’ findings. 

Yes, there was the option ‘do not know’. Despite that, we found a high percentage of missing values in the questions related to knowledge. We checked if the participants who did not answer this knowledge section answered previous ones questions, and in general they did. 

In this paragraph of results, we comment on the percentages of missing values, we did not indicate that the lack of response in these questions is related to the fact that they did not know the answers; we neither do it in the discussion. We appreciate your comment, in this sense we explain this. 

“We did not find differences in social norms and knowledge. For the question: "At least one vaccine on the calendar contains thimerosal", missing values were 17% in the non-hesitant group and 36% in the hesitant group. Similarly, for the question: "The amount of thimerosal in vaccines causes neurotoxicity", missing values were 11.8% and 7.7% in the hesitant and non-hesitant groups, respectively. Similarly, for the question on whether vaccines contain aluminium, 31.8% of values were missing in the hesitant group and 24.7% in the non-hesitant group. Even though there were no statistical differences between both groups. In order to explore better these missing values, we checked if the participants had answered previous questions, and in general they did.” 

Lines 230-235:

These findings are not associated with VH – some degree of causation is suggested in the way this is written. The authors can only present facts i.e. in the vaccine hesitant groups there was a low perception of the severity of whooping cough etc. 

Association is suggesting influence which cannot be determined from this study or questionnaire. 

We appreciate your comment. We agree with your comment about how the factor results are presented and they must not be suggest a causal relation. However, they show an association because the logistic model regression done presents significant values in the probability associated with some factors. It is true that some confidence intervals are wide, and thus, we think that we must detail that the association is slight. 

“The main factors that could be associated with VH were low perception of the severity of whooping cough (aOR: 3.88; 95% CI: 1.32-11.41), low perception of the safety of the HPV vaccine (aOR: 8.50; 95% CI: 1.24-57.8), and the belief that at least one of the vaccines in the current schedule is administered too early (aOR: 6.09; 95% CI: 1.98-18.8) (Table 4). Nurses that did not have children were more likely to report hesitancy (aOR: 4.05; 95% CI:1.22-13.3)” 

Line 234: what is meant by protecting factor?

Thanks for questioning this meaning. The results show that not having children could be an associated factor of VH. Or on the other side, that having children could be a protection against vaccine hesitancy. Therefore, we suggest: 

“Nurses that did not have children were more likely to report hesitancy (aOR: 4.05; 95% CI:1.22-13.3)”

Table 4: I would not present these factors as being associated with VH. These were views of respondents within the VH groups. 

The questionnaire has not addressed the issue of herd immunity and nurses’ views of whether or not this has contributed to illness susceptibility. 

It should be pointed out that the associated factors were just in the study population of paediatric nursing in primary care in Barcelona. We clarify this in the title of table 4, and in other related sentences in the manuscript.

“Table 4. Factors associated with vaccine hesitancy in the paediatric nurses of PHC of Barcelona. 2016-2017.”

In relation to the sentence “the questionnaire has not addressed the issue of herd immunity and nurses’ views of whether or not this has contributed to illness susceptibility”, we did not asked about this two points,…..?¿?

No, the questionnaire didn’t ask about herd immunity.

Table 4: some spelling errors throughout. 

Gender would typically be reported as Male and Female, rather than Man and Woman. 

That is right. Thank you for this comment.

Lines 249-252: sentence is unclear. 

Thanks again for indicating this error, some typing mistake happened.

“The vaccines that generated most doubts were those against HPV, varicella, pneumococcus and hepatitis A. Vaccine-hesitant nurses had a lower perception of the risk for some diseases, a lower perception of the benefit of the varicella and HPV vaccines, and generally more unfavorable beliefs about vaccination (e.g. the time of administration or the number of vaccines) than non-hesitant nurses.”

Line 249 and 254: reference to Catalonia and Barcelona, see comment regarding line 103 and 125. 

Thank you again. We have incorporated this.

“In our study, almost 70% of the paediatric nurses reported acceptance of all the systematic childhood vaccines of the Catalan vaccination schedule in Barcelona and the remaining have questions about the administration of at least one of these vaccines.”

Line 254-257: reference to GPs and other healthcare professionals made. Can the authors refer to studies of nurses as being more relevant?

Referring to studies of nurses would be ideal, both in our countries and in others. However, during the study and during the time of writing this manuscript, we did not find anything more comparable to our target population. Now, again, we have updated a search and,up to our knowledge, there are still missing studies published targeting nurses involved in the pediatric population. Therefore, healthcare professionals in primary care are the most similar population comparable since they share common aims in the care of patients in the community. Obviously, we are aware of the limitations this has. Moreover, although they have different roles in Spain, as in other countries, these healthcare professionals are complementary. Therefore, we are still considering the possibility of comparing our results with the ones that other European countries have published in general practitioners or pediatricians. Finally, we would like to add that this means the lack of knowledge and the need to research about pediatric nursing perception in this area. In addition, we appreciate your comment because we think it is necessary to contextualize these lines and explain to the reader why we are comparing to that. Thus, thank you again.

“Although we would like to compare our results with studies with the same aim targeting, to our knowledge, there are no other published studies with these characteristics, highlighting the research gap in studying VH in this population. We therefore put our findings in the context of studies targeting HCPs (in general) working in PHC. Compared with other European countries, Barcelona paediatric nurses appear to be less likely to have the intention to vaccinate their offspring according to the systematic childhood vaccination schedule than professionals in countries like Switzerland (22), where 95% of paediatricians would vaccinate. On the other hand, general practitioners in France are less likely to recommend vaccines to their patients than to their offspring (32). Other authors, also in France, found that HCPs have divergent immunization attitudes toward their relatives and their patients when asking about the intention of vaccinating their own children(33), especially when considering the newest and most controversial vaccines, like the HPV vaccine. A cross-sectional study done in Croatia through a self-administered questionnaire on attitudes, beliefs and behaviours relating to vaccination among HCPs, including paediatric nurses, reported that nurses were more likely than paediatricians to be vaccine-hesitant (aOR = 5.73, 95%CI = 2.48–13.24). Therefore, in general, our results are similar to those reported across other European studies. (34)”

Line 262: differences in vaccination schedules between regions?

Yes, in Spain there are some small differences in the vaccination schedules between regions. This happened more frequently a decade ago, but since 2012, the central government of Spain, through an Interterritorial Council, proposes a common calendar which sometimes may be slightly modified by each Autonomous Community who has health care competences.

“The vaccines that generate most doubts are those that have been added in the systematic schedule most recently: HPV, introduced in 2008; varicella, in 2016 for infants; and pneumococcus, in 2016. (35) Other authors have commented that poorly communicated changes in the vaccination schedule and changes in scientific understanding can exacerbate feelings of uncertainty [(36)]. In addition, there might be other possible factors influencing doubts in these vaccines (e.g. media, religion, or other socio-political factors). Some authors have seen that changes over time or between regions can motivate HCPs to mistrust in government decisions (36). “ 

Line 264: remove ‘a decade ago’.

Thanks for your advice.

Line 267-268: is this not a more complex issue influenced by social media, misinformation, rather than a scarcity of efficacy studies etc.? This sentence is misleading and potentially inaccurate as it does not portray a more holistic view of all the factors influencing perceptions of HPV vaccine. 

We really appreciate your comment. We agree with your points. Our aim was to express that misinformation led to a perception of ineffectiveness and a lack of confidence in safety. However, it is true that we need to give a broader point of view. In fact, the reference adds (Karafillakis et.al) : “More research needs to be conducted to explore the impact of different types of communication strategies, which would frame the benefits of vaccination as well as risks of not vaccinating. Strategies to better inform public perceptions of vaccines should include the provision of unbiased, comprehensive information tailored to population information needs, and delivered using multiple and new communication technologies such as social media”. Therefore, we propose the next clarification in the sentence.

“This may be related to misinformation in addition to the communication strategies about the infection and the vaccine sometimes portrayed by the media.(37)”

Line 268-271: Again this type of reporting is misleading and damaging if published. I cannot imagine that this was the intention of the authors, however it had a very negative suggestion. 

The conclusion of the authors (Karafillakis et.al) were: “It would be advisable to establish strategies that improve the information that professionals have about HPV, as well as the benefits of the vaccine, so that they transmit them clearly and with assertiveness to parents; this would avoid uncertainty in parents, would improve vaccination rates and decrease the complications of infection (cancer)”.

Moreover, they pointed out in the discussion that “The information that professionals have about the vaccine is not always based on the recommendations of the latest studies'', and “the professionals should have evidence-based information on benefits, efficacy and side effects of the vaccine and health organizations should promote such training”.

We agree with your view. We modify in that sense the lines 268-271:

“The low perception of the safety of HPV vaccine among vaccine hesitant nurses, and not shared with the majority of them, is consistent with a study that found that HCPs in PHC not always receive updated and clear evidence based information about the benefits, efficacy and adverse effects of this vaccine, and the authors see a need of strategies to better inform professionals.[38]”

Line 277-279: Another sentence which could be taken out of context with negative impact. Elaboration of why children who were vaccinated may still contract whooping cough is needed to present a balanced view. 

That is right. In fact, this is an example of a possibility that could erode the perception of effectiveness, but not the perception of severity, that is what the previous lines (273-277) explained. So, we agree with your comment that this sentence may be taken out of context. 

“Different factors have been suggested to contribute to the increase in mild cases of whooping cough since 2010, including among children who had been correctly vaccinated. For example, the evanescence of the protective effect of the pertussis vaccine or the improvement of epidemiology surveillance could have eroded, among other influencing factors, the perception of effectiveness in vaccines in some nurses.[40]”

Line 281: Quite a strong conclusion to make based on two questions which are not a complete reflection of the nurses’ trust in the vaccination schedule; only their views on the two questions that were asked. 

We agree and appreciate your suggestion. Thus the line 281 modification is: 

“A vast majority of paediatric nurses believed that research advances the improvement of vaccines, that vaccines strengthen the immune system, and that they are one of the safest preventive measures. Even though vaccine-hesitant nurses report mistrust in two issues related to the vaccination calendar: they believed that children received more vaccines than needed and that some of them are administered too early.”

Line 288-290: We cannot assume missing values mean the respondent did not know the answer. 

As we indicate in the suggestion proposed for line 222, we do not assume that the missing values mean lack of knowledge. We think that the fact that there is a notable missing values percentage in knowledge, could have made an influence on not finding differences. However, we try to modify the sentence to make it clear and do not cause the reader to think about this relationship.

“Contrary to our assumptions, we observed no differences in knowledge between hesitant and non-hesitant professionals. Both groups showed misconceptions and a moderate proportion of missing values in two questions, which may reflect doubts or information needs.”.

Line 295: which result indicates less than 50% of nurses have enough knowledge on vaccination? What is defined as enough knowledge?

In the questionnaire we included two extra items related to training: 1) if they felt they had enough training and materials to appropriately address families doubts about vaccination and, 2) which type of materials would they prefer to solve doubts of parents (print material, online material, face to face sessions). A 50% of nurses reported having enough information about vaccination. This does not mean that nurses objectively have enough knowledge on vaccination, but that they think they do not have enough training and materials to appropriately manage families' doubts about vaccination. 

“Spanish and Catalan nurse degree training in vaccination should be reinforced, as we found that less than 50% of nurses reported having enough information about vaccination (data not shown). 

Line 298: this claim cannot be made. This is not clearly presented in the results as already indicated in several places in this review, association of factors influencing VH cannot be determined by this questionnaire. 

That is right, continuing with your correction in this sense, we must clarify this point and therefore we present changes in line 298:

“Another important factor that could be associated with VH was not having children. The results suggest that nurses that have children were less likely to be vaccine hesitant. A possible interpretation could be that participants that do not have children may assess the risk of these diseases from a hypothetical standpoint, which is reassessed with better information when having children. It would be interesting to study in more depth the relationship between parenthood and a more favourable attitude to vaccination.”

Line 305-306: Is there a reference or evidence to substantiate this claim?”

It is an author input based on the previous study of Camila Picchio et al. 2019 and Karafillakis et al. 2017 that note that HPV being a new vaccine could explain it VH. Moreover, when we are analysing cross-sectional studies, it is not rare to find selection bias. It could happen that some people prefer not saying their opinion or perception about a topic. This may bias the estimate effect.

“This study has some limitations. Participation was voluntary, which could lead to a selection bias although the percentage of participants was high (>80%). Individuals who did not participate might likely be even more VH. This may modify the estimate effect such a underestimate the magnitude of the problem of the HPV infection.[17, 37]”

Line 306: First time that face to face administration of questionnaire is mentioned – lack of detail in the methods regarding data collection. 

That is right. Considering that the description of the questionnaire was covered with reference Camila Picchio et.al (2019); we did not include this detail. Now, after reading your comments, we will introduce a clear description on the development and use of the questionnaire during the study. Be aware that as long as we have modified this in methods now in this line 306, the term face to face will have more sense. 

“The partial face-to-face administration of the questionnaires could introduce a complacency bias in the responses”.

Line 308: I would determine that 2016-17 (over 11 months) is not a specific period of time but a long period of time. 

We appreciate your suggestion and proposed change instead of “specific period”. We modify it for the long period of time:

“A further limitation is that data were collected for a long period of time, 2016-17, so our results on VH, which is known to vary according to place, time, and context, [35]…”

Line 312: what is CAP? Expand. 

Thank you for this point. It was an error from the Spanish denomination to a primary care center. Therefore, we change it and be consistent with the rest of the manuscript. In addition, we have decided to use an abbreviation for primary health care: PHC; therefore we substitute it here too. 

“…this study accessed all the pediatric nurses of the PHC centers in Barcelona and allowed them to assess the status of a complex multifactorial phenomenon…”

Line 313-315; Again, a leading statement which is not sufficiently backed up by the method or results. 

We appreciate the clarification of this type of corrections.

“Our results highlight four factors that may be associated with VH, and used to compare our context with other populations, and that should be addressed.”

1) I have a concern regarding the focus on issues of vaccines mistrust and negative views in the results and discussion. The paper draws on many references for negative views of vaccinations without a balanced view of trust and positive findings from similarly conducted studies.

2) I also have concerns regarding the questionnaire development which draws on several health behaviour theories. 

3) The confidence intervals and odds ratios do not appear to back up the strength of findings posed by the authors. The results do not support the conclusions sufficiently. 

1) According to the awareness of the negative view highlighted in the manuscript we included some new sentences in the manuscript that pretend to balance the view. We agree with the relevance to show what professionals adequately perceive, know, and believe about pediatric vaccination. 

About perception of risk and benefit, we consider to add : 

“In more than half of the vaccines, the perception of the risk of infection is similar in non-hesitancy nurses and hesitancy nurses that have been asked. Nursing professionals who were hesitant to vaccinate had a low perceived risk of infection for 5 of the 14 diseases in the vaccination schedule…”

About beliefs, we will like to add some more positive result in lines 220:

“Moreover, almost all nurses participating agreed that vaccines strengthen the immune system (88,1% VH; 92,2% non-VH)”.

In the discussion: 

“Overall, the findings revealed that most nurses in PHC have a positive perception of childhood vaccination. In our study, almost 70% of the paediatric nurses reported acceptance of all the systematic childhood vaccines of the Catalan vaccination schedule in Barcelona and the remaining have questions about the administration of at least one of these vaccines. The vaccines that generated most doubts were those against HPV, varicella, pneumococcus and hepatitis A. Vaccine-hesitant nurses had a lower perception of risk caused by some diseases, a lower perception of the benefit of the varicella and HPV vaccines, and generally more unfavourable beliefs about vaccination (e.g. the time of administration or the number of vaccines) than non-hesitant nurses.

Although we would like to compare our results with studies with the same aim targeting, to our knowledge, there are no other published studies with these characteristics, highlighting the research gap in studying VH in this population. We therefore put our findings in the context of studies targeting HCPs (in general) working in PHC. Compared with other European countries, Barcelona paediatric nurses appear to be less likely to have the intention to vaccinate their offspring according to the systematic childhood vaccination schedule than professionals in countries like Switzerland (22), where 95% of paediatricians would vaccinate. On the other hand, general practitioners in France are less likely to recommend vaccines to their patients than to their offspring (33). Other authors, also in France, found that HCPs have divergent immunization attitudes toward their relatives and their patients when asking about the intention of vaccinating their own children(34), especially when considering the newest and most controversial vaccines, like the HPV vaccine. A cross-sectional study done in Croatia through a self-administered questionnaire on attitudes, beliefs and behaviours relating to vaccination among HCPs, including paediatric nurses, reported that nurses were more likely than paediatricians to be vaccine-hesitant (aOR = 5.73, 95%CI = 2.48–13.24). Therefore, in general, our results are similar to those reported across other European studies.”

“A vast majority of paediatric nurses believed that research advances the improvement of vaccines, that vaccines strengthen the immune system, and that they are one of the safest preventive measures. Even though vaccine-hesitant nurses report mistrust in two issues related to the vaccination calendar: they believed that children received more vaccines than needed and, that some of them are administered too early.These aspects suggest some gaps in the knowledge about the reasons and timing of vaccination. Moreover, we have found an association between the distrust in the pharmaceutical industry, although not with government health authorities, and the nurses’ perception about vaccine administration timing. Another study observed that the perceived lack of transparency of administrations could lead to mistrust about changes in the vaccination calendar. [10]”

2) In relation to the questionnaire development, we hope that now with the proper explanation and the inclusion of a more precise description of the data collection this point could be solved. 

3) It is true that we might make strong conclusions about associated factors from some results that have a wide confidence interval, and this does not allow us to conclude with full security that relation. Here, again, we hope that we have edited these points and now, the reader could understand better the results and our interpretation. 

6. PLOS authors have the option to publish the peer review history of their article (what does this mean?). If published, this will include your full peer review and any attached files.

Do you want your identity to be public for this peer review? For information about this choice, including consent withdrawal, please see our Privacy Policy.

Reviewer #1: No

Reviewer #2: No

We checked our figure through Preflight Analysis and Conversion Engine (PACE) digital diagnostic tool.

Thank you again for all the constructive reflexion and for giving us the opportunity to submit a revised draft of our manuscript. 

We look forward to hearing from you in due time regarding our submission and to respond to any further questions and comments you may have.

Sincerely,

Usue Elizondo-Alzola

22nd of April 2021

---

## [Editor Report · Decision Letter 1]

3 May 2021

Vaccine hesitancy among paediatric nurses: Prevalence and associated factors

PONE-D-20-36118R1

Dear Dr. Elizondo Alzola,

We’re pleased to inform you that your manuscript has been judged scientifically suitable for publication and will be formally accepted for publication once it meets all outstanding technical requirements.

Kind regards,

Ray Borrow, Ph.D., FRCPath

Academic Editor

PLOS ONE
---

## [Editor Report · Acceptance letter]

10 May 2021

PONE-D-20-36118R1 

Vaccine hesitancy among paediatric nurses: Prevalence and associated factors 

Dear Dr. Elizondo-Alzola:

I'm pleased to inform you that your manuscript has been deemed suitable for publication in PLOS ONE. Congratulations! Your manuscript is now with our production department. 

Kind regards, 

on behalf of

Prof. Ray Borrow 

Academic Editor

PLOS ONE